



# A new Description of Probability Density Distributions of Polar Mesospheric Clouds (PMC)

Uwe Berger[1], Gerd Baumgarten[1], Jens Fiedler[1], and Franz-Josef Lübken[1]

[1]Leibniz-Institute of Atmospheric Physics, Rostock University, Kühlungsborn, Germany

**Correspondence:** Uwe Berger (berger@iap-kborn.de)

**Abstract.** In this paper we present a new description about statistical probability density distributions (pdfs) of Polar Mesospheric Clouds (PMC) and noctilucent clouds (NLC). The analysis is based on observations of maximum backscatter, ice mass density, ice particle radius, and number density of ice particles measured by the ALOMAR RMR-lidar for all NLC seasons from 2002 to 2016. From this data set we derive a new class of pdfs that describe the statistics of PMC/NLC events which is different from previously statistical methods using the approach of an exponential distribution commonly named g-distribution. The new analysis describes successfully the probability statistic of ALOMAR lidar data. It turns out that the former g-function description is a special case of our new approach. In general the new statistical function can be applied to many kinds of different PMC parameters, e.g. maximum backscatter, integrated backscatter, ice mass density, ice water content, ice particle radius, ice particle number density or albedo measured by satellites. As a main advantage the new method allows to connect different observational PMC distributions of lidar, and satellite data, and also to compare with distributions from ice model studies. In particular, the statistical distributions of different ice parameters can be compared with each other on the basis of a common assessment that facilitate, for example, trend analysis of PMC/NLC.

## 1 Introduction

First studies of probability distributions of Polar Mesospheric Clouds (PMC) and noctilucent clouds (NLC) were reported by *Thomas* (1995) using data from the UVS instrument on board the Solar Mesosphere Explorer (SME) satellite and from the Solar Backscatter Ultraviolet (SBUV) instrument on the Nimbus-7 satellite over the period 1978-1986, measuring scattered limb albedo at 265 nm and nadir albedo at 273.5 nm, respectively. *Thomas* (1995) introduced empirical measures in the statistical analysis of PMC brightness distributions. He showed that the frequency distribution of PMC albedo derived from both SME and SBUV satellite data can be approximated by a (normalized) exponential probability function, see Figure 3 in *Thomas* (1995). Secondly, the author also proposed to use cumulative frequency numbers (the so-called g-function) of clouds $g(A)$ exceeding a certain albedo $A$, in order to better represent the exponential populations. Examples of g-distributions are





plotted on a semi-logarithmic scale in Figure 4 in *Thomas* (1995), clearly indicating an approximately linear behavior of cumulative frequencies in a logarithmic format.

In the following years many observational PMC analysis of seasonal statistics have been published frequently using the g-function, e.g. reports from Wind Imaging Interferometer (WINDII) and Polar Ozone and Aerosol Measurement II data (*Shettle*
*et al.*, 2002), SBUV data (*Deland et al.*, 2003), Student Nitric Oxide Explorer (SNOE) data (*Bailey et al.*, 2007), ice water content data derived from SBUV (*DeLand and Thomas*, 2015), or ALOMAR lidar data (*Fiedler et al.*, 2017). Also model analysis have used the g-function investigating trends and long-term changes in PMC parameters (*Lübken et al.*, 2013; *Berger and Lübken*, 2015).

The g-function approach has been relatively successfully applied to many kinds of different PMC parameters as brightness,
albedo, maximum backscatter ratio, integrated backscatter, ice water content, ice mass densities, ice particle size, or ice particle number density since frequency histograms of all these parameters have sometimes a nearly, at least piecewise exponential shape. Furthermore, sometimes PMC data seem to fit almost perfectly to exponential distributions, particularly when using cumulative standardizations of data (*Thomas*, 1995). An example of a good exponential fit is the frequency distribution of ALOMAR backscatter data that are discussed in Sect. 3.1.1. On the other hand, in some statistical applications it is obvious
that the exponential approach describes the data rather insufficiently, see examples of ice mass density, ice radius and ice number density in Sect. 3.1.2. Therefore it is a desirable task to provide some more aspects on the theory of PMC/NLC statistic.

This paper makes an attempt to investigate in more detail the statistic of probability density functions (pdf) of PMC/NLC climatology for various ice parameters. In the following we analyze a PMC/NLC data record of maximum backscatter, ice mass
density, ice particle radius, and number density from the period 2002 – 2016 measured by the ALOMAR RMR lidar. From the analysis of these ALOMAR data, we derive a new class of pdfs of PMC/NLC distributions that, as we will show, modifies and improves the exponential (g-function) approach as introduced by *Thomas* (1995).

## 2   Discription of ALOMAR lidar data

The data set obtained by the ground-based Rayleigh/Mie/Raman (RMR)-lidar, located at the Arctic station ALOMAR (69°N,
16°E), consists of occurrence frequency, brightness and altitude of NLCs (noctilucent clouds). The RMR-lidar is in operation on a routine basis during the summer seasons (NLC season: 20 May to 20 August) since 1997. Since summer 2002 the lidar system has the general capability to run in a multiple wavelength (3-color) mode. We shortly summarize the 3-color lidar technique: laser pulses at three separated wavelengths (355 nm, 532 nm, 1064 nm) are emitted, scattered back by air molecules and ice particles in the atmosphere and collected by telescopes. The received light is recorded by single photon counting
detectors. After separation of the ice particle and molecular backscatter signal, we extract three vertical profiles of so-called backscatter ratios which are a measure of height dependent brightness of the ice cloud. From each backscatter height profile we estimate a maximum backscatter (MBS) signal which corresponds to mean height of maximum brightness as an average over the three wavelengths that is typically located at an altitude near 83 km. We assume that at the altitude of MBS the actual





shape of the ice particle distribution can be described by a Normal-distribution. Then we derive from the three measured colors the characteristics of the Normal-distribution with mean ice radius, ice number density and variance. Finally, we also estimate from these values the actual ice mass density (IMD) at MBS heights. Such a Gaussian assumption has been widely used in PMC/NLC data processing of lidar and satellite data, e.g. ALOMAR lidar (*Baumgarten et al.*, 2010) and AIM satellite with

SOFIE/CIPS instruments (*Hervig and Stevens*, 2014; *Bailey et al.*, 2015). Also microphysical model studies show a strong evidence of Gaussian distributed ice particles at the height of maximum brightness of PMCs, e.g. *Berger and von Zahn* (2002), *Rapp and Thomas* (2006).

In this paper we will analyze the climatology of all ice seasons from 2002 until 2016 merging all 15 seasons to one data record. Within this combined data set we then get a total number $N$ of 8,597 observations which is sufficiently numerous in

order to avoid too large statistical irregularities in a frequency histogram of the data.

## 3    The exponential probability distribution (g-function)

In general, the seasonal climatology of PMC events with measured ice parameters as e.g. integrated backscatter, maximum backscatter, column ice mass, albedo or ice mass density, has been supposed to follow an exponential distribution which we name $\mathcal{E}(x)$ with ice parameter variable $x$. In the following we summarize the general characteristics of the exponential distri-

bution which allows to compute a numerical test for exponential distributed data. The properties of the exponential probability distribution will be also compared with the characteristics of our new probability distribution approach introduced in Sect. 4.

The general form of the exponential distribution $\mathcal{E}(x)$ with scale parameter $\alpha > 0$ is defined as a probability density function (pdf) given by $\mathcal{E}(x) = \alpha \exp(-\alpha x)$ which fulfills the normalization condition of a pdf with $\int_0^\infty \mathcal{E}(x)dx = 1$. *Thomas* (1995) defined the g-function $g(x)$ as the cumulative probability $\mathcal{E}_{cum}$ with

$$g(x) = \mathcal{E}_{cum}(x) = \int_x^\infty \alpha e^{-\alpha x} \, dx' = e^{-\alpha x} \; . \tag{1}$$

Taking the logarithm of $\mathcal{E}$ yields a straight line $\ln(\mathcal{E}) = \ln \alpha - \alpha x$. For a given class of values $[x_1; x_2]$ the likeliness of this class is proportional to the area enclosed by the continuous probability distribution and is obtained by integrating $\mathcal{E}$ on the segment length (bin size) $\Delta x = x_2 - x_1$ as $\int_{x_1}^{x_2} \mathcal{E}dx = -e^{-\alpha x_2} + e^{-\alpha x_1}$.

A statistical analysis of ice parameters has to take into account the aspect of specific sensitivities of different instruments. For

example the ALOMAR lidar is generally sensitive to a backscatter signal larger than a threshold about $2 - 3 \cdot 10^{-10} m^{-1} sr^{-1}$ (*Fiedler et al.*, 2017). When considering a threshold $(x_{th})$ the exponential pdf $\mathcal{E}(x)$ is normalized according to $A \int_{x_{th}}^\infty \mathcal{E}(x)dx = 1$ with a scaling factor $A = \exp(\alpha x_{th})$. We summarize the properties of the exponential distribution taking into account a threshold in Appendix A.

For a threshold of zero $(x_{th} = 0)$ we get the regular exponential distribution $\mathcal{E}(x)$ which has the mean $\mu = 1/\alpha$, median

$\nu = \ln(2)/\alpha$, mode $\eta = 0$, variance $\sigma^2 = 1/\alpha^2$ and standard deviation $\sigma = 1/\alpha$. Note that the exponential distribution has the unique property that the mean $\mu$ and standard deviation $\sigma$ are identical, see also Eq. (A1) and Eq. (A4). In combination with





the median (Eq. A2), these equations form a simple statistical constraint, namely

$$\mu - x_{th} = \sigma = (\nu - x_{th})/\ln(2) \,. \tag{2}$$

This allows to test whether a given observational data sample shows good conformity with an exponential (g-function) distribution.

For a given data sample $x_i$ ($i = 1, ..., N$) assuming a threshold $x_i > x_{th}$ we use the common estimates of mean $\bar{m}$ and variance $s^2$ (standard deviation $s$) with

$$\bar{m} = \frac{1}{N} \sum_i^N x_i \quad , \quad s^2 = \frac{1}{N-1} \sum_i^N (x_i - \bar{m})^2 \quad , \quad x > x_{th} \,. \tag{3}$$

In addition we also calculate the median $\tilde{m}$ and mode $\hat{m}$ which is the value that occurs most frequently in the data sample. Hence testing a data sample to be exponentially distributed means that mean, median and standard deviation of the sample have to fulfill the following identity:

$$\mu - x_{th} = \sigma = \frac{\nu - x_{th}}{\ln(2)} \quad \longrightarrow \quad \bar{m} - x_{th} = s = \frac{\tilde{m} - x_{th}}{\ln(2)} \,. \tag{4}$$

We will use this condition to analyze the ALOMAR data with respect to possible exponential (g-function) distributions.

### 3.1 Analysis of ALOMAR data on exponential distributions (g-function)

#### 3.1.1 Analysis of maximum backscatter data

We investigate the frequency distribution of maximum backscatter (MBS) data in units of $10^{-10} m^{-1} sr^{-1}$. We assume a threshold of 3 that corresponds to the instrumental sensitivity of the ALOMAR lidar. Then we sort the $x$=MBS data to a bin size of one per class starting from the threshold value and calculate a frequency histogram. Finally, we normalize the histogram so that the sum of all frequency classes equals one.

Figure 1a shows the frequency distribution of x=MBS data in a semi-logarithm scale. The first impression is that the data points are almost perfectly approximated by a linear regression besides some statistical noise. This indicates that a exponential function describes the distribution of data with a high accuracy. Figure 1b shows the distribution histogram in a original non-logarithmic representation. We see that the exponential fit matches the data histogram with a high precision. Consequently, the relative error is rather small (6.5 %). The high quality of the fit is also supported by the fact that theoretical mean, median, mode, and standard deviation ($\mu$, $\nu$, $\eta$, $\sigma$) using Eq. (A1–A4) and estimates of mean, median, mode, and standard deviation ($\bar{m}$, $\tilde{m}$, $\hat{m}$, $s$) from the data sample derived from Eq. (3) all coincide within their error bars. Now we perform the proposed exponential (g-function) test with $\bar{m} - x_{th} = s = (\tilde{m} - x_{th})/\ln(2)$, see Eq. (4), and insert the values from the data sample of mean ($\bar{m} = 12.0 \pm 0.3$), median ($\tilde{m} = 9.0 \pm 0.4$), and standard deviation ($s = 9.2 \pm 0.5$). The error uncertainties have been estimated with bootstrap methods. We find that $\bar{m} - x_{th} = 12.0 - 3 = 9 \pm 0.3$, $s = 9.2 \pm 0.5$, and $(\tilde{m} - x_{th})/\ln(2) = (9.0 - 3)/0.69315 = 8.7 \pm 0.6$. Hence the identity is fulfilled when allowing for uncertainties introduced by statistical errors. We conclude that lidar MBS-data are very likely exponentially distributed and follow a g-function, respectively.



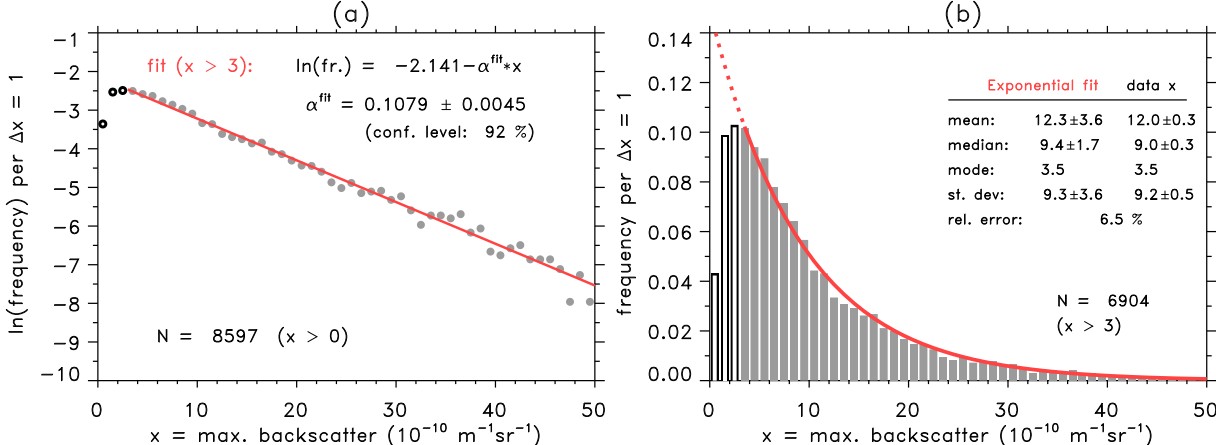

**Figure 1.** (a) Logarithm of frequency distribution of maximum backscatter (x=MBS) in units of $10^{-10}\ m^{-1}sr^{-1}$ (gray points $x > 3$; black circles $0 < x < 3$). The bin size is $\Delta x = 1$. The straight line (solid red) has been derived from a least squares fit to MBS data with $x > 3$. (b) Same as panel a, but original, non-logarithmic frequency distribution (gray bars $x > 3$; black bars $0 < x < 3$). The exponential fit derived from panel a is shown as a red curve. Values of mean, median, standard deviation are given to compare fit and original data taking into account a threshold of $x_{th} = 3$. The relative error given in percent describes the quality of exponential fitting.

### 3.1.2 Analysis of ice mass density, ice radius, and ice number density data

Now we investigate other ice parameters from the ALOMAR data set with respect to exponential distributions, namely the frequency distributions of ice mass density (IMD) in units of mg $\cdot$ m$^{-3}$ (threshold 20, bin size of 2), ice radius $r$ in units of nm (threshold 20, bin size of 1) and ice number density $n$ in units of cm$^{-3}$ (threshold 30, bin size of 10). We will show that these parameters do not follow an exponential distribution (g-function) as expected. In Figure 2a we plot the frequency distribution for $y =$ IMD data in a semi-logarithmic scale. Obviously, the data points have no dominant linear shape. There exist systematic deviations between data and theoretical exponential fit. In comparison to the fit curve, data points are systematically smaller at $y = 20 - 40$. Vice versa, data points exceed substantially fit values in the range $y = 40 - 90$. Also, frequencies in all classes below the threshold are significant smaller than a proposed exponential fit. Indeed, the frequency histogram in the non-logarithmic frame shows these systematic deviations between data and exponential fit even more pronounced, see Figure 2b. With a relative error of about 19 % the exponential curve fails to fit satisfactorily the data. Also significant differences exist between fit and data parameter of mean, median, mode, and standard deviation (Figure 2b). Finally, we apply the exponential (g-function) test for IMD data and get the following results: finding the mean ($\bar{m} = 62.5 \pm 1.3$), median ($\tilde{m} = 53.5 \pm 1.4$), and standard deviation ($s = 35.2 \pm 1.2$) directly calculated from the data sample, we get $\bar{m} - y_{th} = 62.5 - 20 = 42.5$ unequal $s = 35.2$ unequal $(\tilde{m} - y_{th})/\ln(2) = (53.5 - 20)/0.69315 = 48.3$. Hence the condition of identity is not satisfied even allowing for uncertainties introduced by statistical errors again calculated from bootstrap methods. That is why we have to conclude that the lidar IMD data are very likely not exponentially distributed. When we investigate a possible exponential distribution for ice radius $r$ and ice number density $n$ data, see Figure 2c–f, we even see larger discrepancies between data and exponential





**Figure 2.** (a) Logarithm of frequency distribution of ice mass density (y=IMD) in units of mg/m$^3$ (gray points $y > 20$; black circles $0 < y < 20$). The bin size is $\Delta y = 2$. The straight line (solid red) has been derived from a least square fit to IMD data with $y > 20$. (b) Same as panel a, but original, non-logarithmic frequency distribution (gray bars $y > 20$; black bars $0 < y < 20$). The exponential fit derived from panel a is shown as a red curve. Values of mean, median, standard deviation are given to compare fit and original data taking into account a threshold of $y_{th} = 20$. The relative error given in percent describes the quality of exponential fitting. (c) and (d) Same, but for ice radius $r$ in units of nm with bin size $\Delta r = 1$ and threshold $r_{th} = 20$. (e) and (f) Same, but for ice number density $n$ in units of 1/cm$^3$ with bin size $\Delta n = 10$ and threshold $n_{th} = 30$.





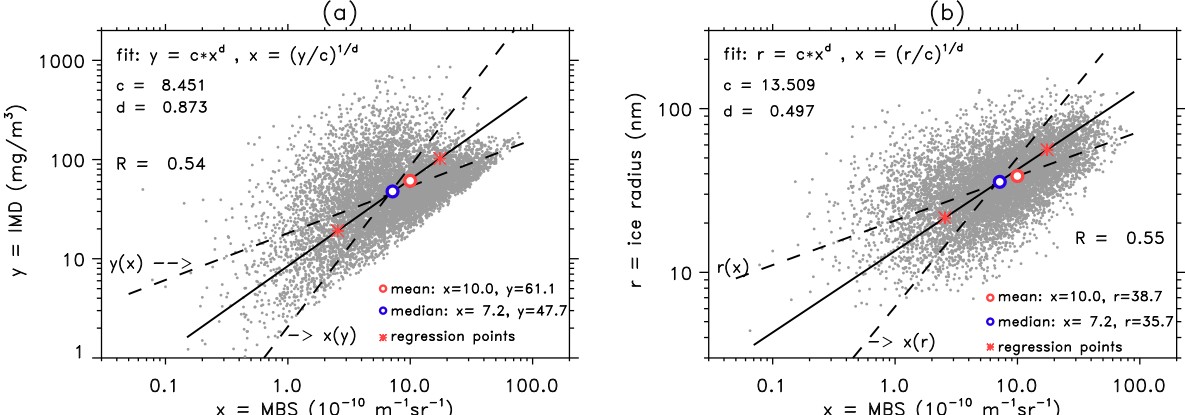

**Figure 3.** (a) Maximum backscatter ($x$=MBS) versus ice mass density ($y$=IMD) in a logarithmic frame for all data with correlation coefficient $R$ and regression parameters $c$ and $d$, see text for more details. Regression points $p_{1/2} = [m \pm \Delta m; n \pm \Delta n]$ are calculated with $m = 1/N \sum_i^N \ln x_i$, $n = 1/N \sum_i^N \ln y_i$, $\Delta m = \sqrt{1/N \sum_i^N (\ln x_i - m)^2}$, and $\Delta n = \sqrt{1/N \sum_i^N (\ln y_i - n)^2}$. Mean and median are calculated from original, non-logarithmic data. (b) Same for $x$=MBS versus ice radius $r$.

fits with e.g. relative errors about 29 %, indicating that also both $r$ and $n$ are very likely not exponentially distributed. This is supported by the fact that the test of mean, median, and variance fails again and shows large inequalities.

We summarize that ice mass density, ice radius, and ice number density do not follow an exponential distribution in contrast to maximum backscatter. In the following section we will show that this is reasonable and is based on the fact that a functional

5 link between MBS to the other data sets of IMD, $r$ and $n$ does miss a linear relationship.

### 3.2  Test on linearity between maximum backscatter and ice mass density, ice radius, ice number density data

Linearity between maximum backscatter (MBS) and ice mass density (IMD), ice radius $r$ and ice number density $n$ data is a necessary and sufficient condition that also IMD, $r$ and $n$ data samples are exponentially distributed, see also next section. In the following we will test this constraint. Figure 3a shows a scatter plot in a logarithmic frame for simultaneously measured

10 MBS and IMD data. In order to test a linear relationship between x=MBS and y=IMD we introduce a general fit function described by a power law condition as

$$y(x) = cx^d \iff x(y) = (y/c)^{1/d} \tag{5}$$

with the two constants $c$ (linear constant) and $d$ (power constant). Only for $d = 1$ we expect a perfect linear dependence between $x$ and $y$. First of all, the logarithmic values of $x$ and $y$ do not yet have a high linear correlation ($R = 0.54$), see Figure 3a. Since

15 the correlation coefficient $R$ is unequal one, the two regression lines resulting from $y(x): \ln x \mapsto \ln y$ and $x(y): \ln y \mapsto \ln x$ differ from each other. This means that the best choice of a regression fit is determined by a straight line through the two regression points which are defined by the means plus/minus standard deviations of logarithmic $x$ and $y$ data. Note that the





positions of regression points also relate to the half width of the angle which is spanned by the two regression lines $y(x)$ and $x(y)$. For our mean regression line we estimate $d = 0.873$. The statistical error for $d$ is $\Delta d = \pm 0.012$ with a confidence level of 95 % which indicates a significant non-linearity. Hence we conclude that the pdf describing the distribution of IMD data is very likely not an exact exponential function and its cumulative distribution does not follow precisely a g-function description

because the criteria of 'linearity' is violated. In Figure 3b we show a second example for the correlation between MBS and ice radius $n$. Again the correlation is about $R = 0.55$, but now the power constant is even much smaller with $d = 0.497$ which is far away from unity. Finally, we investigated the linearity between MBS and ice number density $n$ where we find a weak negative correlation of $R = -0.15$ (not shown here). A best fit analysis yields a power value of $d = -0.534$ which again fails significantly the constraint of unity. Hence we conclude that also ice radius and ice number densities distributions should not

follow exponential (g-function) distributions.

## 4 A new probability density function for PMC parameters

In this section we will present the major part of the new statistical approach in order to describe frequency distributions of different PMC/NLC parameters.

There exists a general mathematical method ('integration by substitution') that provides the opportunity to transform between

probability density functions (pdf) with different statistical variables. This is done by the following procedure: Assuming a given pdf $P(x)$ with variable $x$, then the transformation from $x$ to a new variable $y(x)$ with a new pdf $Q(y)$ is specified by

$$Q(y) \; = \; \|\partial x / \partial y\| \cdot P(x(y)) \tag{6}$$

with $x(y)$ being the inverse function of $y(x)$. Here the absolute value of the derivative $\partial x / \partial y$ has to be calculated so that the new pdf $Q$ is defined positively everywhere. In order to apply this approach one needs generally two requirements: (1) Any

transformation between the two pdfs $P$ and $Q$ needs an initial guess in one of the two pdfs, either $P$ or $Q$. (2) An analytic formula of a forward and backward model must be available that describes the functional dependence between the two statistic ice variables $x$ and $y$. In the following we discuss how we satisfy these two requirements.

We apply this method for two ice parameters, namely MBS with variable $x$ and an unknown ice parameter named $u$ (e.g. this unknown ice parameter might be ice particle radius). For condition (1) we use the hypothesis that the distribution of maximum

backscatter data (MBS) is perfectly represented by an exponential pdf and its cumulative distribution is described by a g-function according to Eq. (1). For condition (2) we assume a power form of a fit function used in Eq. (5) that also allows to calculate analytically the inverse function. We discuss a suitable justification of this assumption in Sect. 6.2. Hence the forward model is $u(x) = cx^d$ and the backward model is $x(u) = (u/c)^{1/d}$. Then the new distribution $\mathcal{U}$ for the arbitrary ice parameter $u$ using Eq. (6) is given by

$$\mathcal{U}(u) \; = \; \|\partial x / \partial u\| \cdot \mathcal{E}(x(u)) = \| \frac{1}{du} \cdot \left( \frac{u}{c} \right)^{1/d} \| \cdot \alpha \, e^{-\alpha (u/c)^{1/d}} \; . \tag{7}$$


Equation (7) can be simplified to a more general form with

$$\mathcal{U}(u) \;=\; \tilde{a}|\tilde{b}|u^{\tilde{b}-1}\mathrm{e}^{-\tilde{a}u^{\tilde{b}}} \;\;,\;\; \tilde{a} = \alpha\left(\frac{1}{c}\right)^{1/d} \;\;,\;\; \tilde{b} = \frac{1}{d}\;. \tag{8}$$

In a next step we introduce in an arbitrary manner a third ice parameter named $z$, for which we assume again the same power law (Eq. 5) now valid between $z$ and $u$ as

$$z(u) \;=\; \tilde{c}u^{\tilde{d}} \;\Leftrightarrow\; u(z) \;=\; (z/\tilde{c})^{1/\tilde{d}}\;.$$

Again we apply Eq. (6) and calculate the unknown pdf $\mathcal{Z}(z)$:

$$
\begin{aligned}
\mathcal{Z}(z) \;=\; \|\partial u/\partial z\| \cdot \mathcal{U}(u(z)) \;=&\; \|\frac{1}{\tilde{d}z}\cdot\left(\frac{z}{\tilde{c}}\right)^{1/\tilde{d}}\|\cdot \tilde{a}\tilde{b}\left((z/\tilde{c})^{1/\tilde{d}}\right)^{\tilde{b}-1}\mathrm{e}^{-\tilde{a}(z/\tilde{c})^{\tilde{b}/\tilde{d}}} \\
\;=&\; \|\frac{\tilde{a}\tilde{b}}{\tilde{d}}\|\cdot z^{-1}\left((z/\tilde{c})^{1/\tilde{d}}\right)^{\tilde{b}}\mathrm{e}^{-\tilde{a}(z/\tilde{c})^{\tilde{b}/\tilde{d}}}\;.
\end{aligned}
$$

At first glance the algebraic expression for $\mathcal{Z}$ looks particulary complex, but $\mathcal{Z}$ can be transformed to a general form with

$a = \tilde{a}(1/\tilde{c})^{\tilde{b}/\tilde{d}}$ and $b = \tilde{b}/\tilde{d}$ as

$$\mathcal{Z}(z) \;=\; a|b|z^{b-1}\mathrm{e}^{-az^{b}} \quad (a>0,\, b\neq 0)\;. \tag{9}$$

Equation (9) represents our final result. The pdf $\mathcal{Z}(z)$ describes the general form of the new statistical distribution. Note that the algebraic expressions of Eq. (8) and Eq. (9) formally coincide. This means that any probability distribution of a new ice parameter that is connected to other ice parameters through our functional power law (Eq. 5), can be described by the general

pdf given by Eq. (9). The constants $a$ and $b$ represent two free parameters in the $\mathcal{Z}$-distribution which we name the scale parameter $a$ and the shape parameter $b$. Obviously, the $\mathcal{Z}$-pdf is identical with an exponential pdf (or g-function) in the limit $b = 1$. This shows the close interconnection of the new $\mathcal{Z}$-pdf to the commonly used exponential (g-function) approach. We will show in the following that any distribution from so different ice parameters as maximum backscatter, ice mass density, ice radius, and number density of ice particles can be described on a uniform basis with a high accuracy by $\mathcal{Z}$. Vice versa this

indicates that these ice parameters are connected depending on each other by the uniform power law relation (Eq. 5), more details are discussed in Sect. 6.2.

## 5 Application of the $\mathcal{Z}$-distribution to real data

### 5.1 General properties of the $\mathcal{Z}$-distribution

In this section we first show some general characteristics of the new $\mathcal{Z}$-distribution. From these properties we derive conditions

and constraints that will allow to estimate the specific values of the two free constants in $\mathcal{Z}$, the scale parameter $a$ and shape parameter $b$, for a given data sample.

First we show that $\mathcal{Z}$ is a correct pdf satisfying the normalization condition $\int\limits_{0}^{\infty}\mathcal{Z}dz = 1$:

$$\int\limits_{0}^{\infty}\mathcal{Z}dz \;=\; \int\limits_{0}^{\infty}a|b|z^{b-1}\mathrm{e}^{-az^{b}}dz \;=\; -\frac{|b|}{b}\mathrm{e}^{-az^{b}} + C = 1\;.$$



The definition range of $\mathcal{Z}(z;a,b)$ is $z \geq 0$, $a > 0$, and $b \neq 0$ with $\mathcal{Z}(z < 0) = 0$ and

$$\mathcal{Z} = a|b|z^{b-1}\mathrm{e}^{-az^b} \ , \quad \ln(\mathcal{Z}) = \ln(a|b|) + (b-1)\cdot\ln(z) - az^b \ , \quad b > 0 \ . \tag{10a}$$

For a negative $b$ the distribution $\mathcal{Z}$ is described by

$$\mathcal{Z} = a|b|\left(\frac{1}{z}\right)^{|b-1|}\mathrm{e}^{-a(1/z)^{|b|}} \ , \quad \ln(\mathcal{Z}) = \ln(a|b|) + |b-1|\cdot\ln(1/z) - a(1/z)^{|b|} \ , \quad b < 0 \ . \tag{10b}$$

The cumulative form of $\mathcal{Z}$ for $b > 0$ is given by

$$\mathcal{Z}_{cum}(z) = \int_z^\infty \mathcal{Z}dz' = \mathrm{e}^{-az^b} \ , \quad \ln(\mathcal{Z}_{cum}) = -az^b \ , \quad \ln(|\ln(\mathcal{Z}_{cum})|) = \ln(a) + b\ln(z) \ . \tag{11a}$$

For $b < 0$ we have to choose the cumulative calculation in reverse order starting the integration at zero. Naming the reverse cumulative with index zero as $\mathcal{Z}_{cum}^0$ we get

$$\mathcal{Z}_{cum}^0(z) = \int_0^z \mathcal{Z}dz' = \mathrm{e}^{-a(1/z)^{|b|}} = \mathrm{e}^{-az^b} \ , \quad \ln(\mathcal{Z}_{cum}^0) = -az^b \ , \quad \ln(|\ln(\mathcal{Z}_{cum}^0)|) = \ln(a) + b\ln(z) \ . \tag{11b}$$

Only, the cumulative descriptions from Eq. (11a,b) allow in principle to estimate roughly the constants $a$ and $b$ for a given data sample using the double logarithmic functional dependence, whereas the direct logarithm of $\mathcal{Z}$ (Eq 10a,b) offers no possibility to solve for $a$ and $b$. However, the method calculating the double logarithmic cumulative is not recommended. Several numerical tests showed that a stable estimation of $a$ and $b$ from noisy data applying this double logarithmic approach is an almost impossible task. Instead, we propose two different methods that rely on much more powerful principles (see next

Sect. 5.2). Additionally, we have to take care about a possible negative value of $b$ that can be only identified using Eq. (11b). In fact such a case occurs in the analysis of ALOMAR data. We will give in Sect. 5.3 an example that only a negative slope parameter describes the distribution of number density of ice particles.

Generally, the $\mathcal{Z}$-distribution has the ability to characterize many different types of distributions, see Figure 4. Especially, the shape parameter $b$ determines the shape of the $\mathcal{Z}$-distribution describing non-linear exponential, exponential, right-skewed, left-

skewed or symmetric curves. For $0 < b < 1$ the pdf increase is non-linear exponentially accelerated to infinity as $z$ approaches zero. For $b = 1$ the pdf is exactly an exponential distribution having a positive finite value for $z$ equal zero. For $b > 1$, the function tends to zero as $z$ approaches zero. When $b$ is between one and two, the function is right-skewed and rises to a peak quickly, then decreases for large $z$. When $b$ has approximately a value between three and four, the function becomes symmetric and bell-shaped like a normal distribution. Note that exact symmetry is given for a skewness equal to zero which is true at

$z = 3.60232$. For $b$ values larger than approximately five, the function becomes again asymmetric changing the skewness to the left. For $b < 0$, the function is skewed to the right and decreases steeply towards zero as $z$ approaches zero. Note that $\mathcal{Z}$ is never negative and owns a local maximum described by the mode whenever $b$ is negative or larger than one. Finally we see that a double logarithmic presentation of cumulative functions describes linear shapes with slope $b$, see Figure 4j-l.



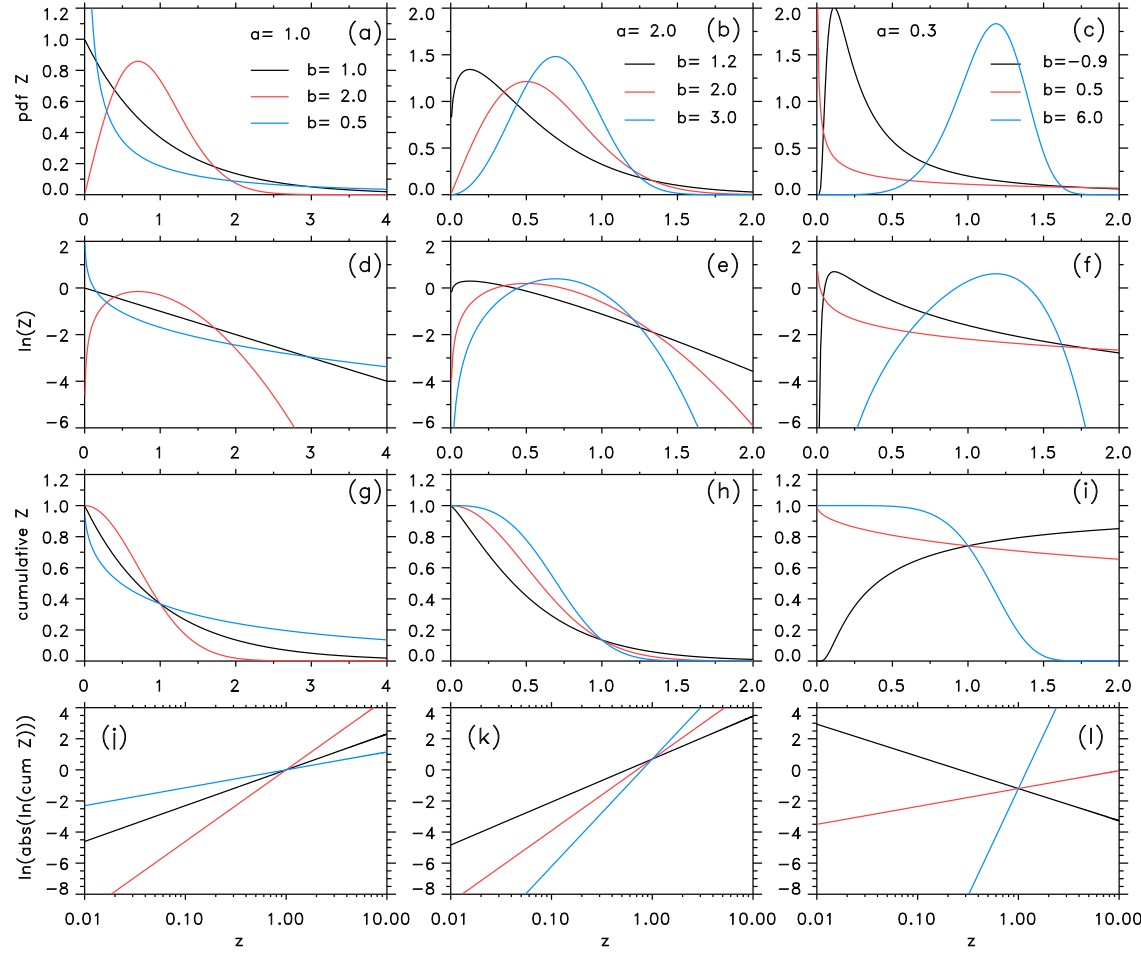

**Figure 4.** (a–c) Examples of $\mathcal{Z}(z)$-function with different parameter values $a$ and $b$, see Eq. (10). (d–f) same but for $\ln(\mathcal{Z})$ from Eq. (10). (g–i) Same but for $\mathcal{Z}_{cum}$ from Eq. (11). (j–l) Same but for $\ln(|\ln(\mathcal{Z}_{cum})|)$ from Eq. (11).

It is interesting to note that our new $\mathcal{Z}$-distribution is closely related to a more general Weibull-distribution (*Wilks*, 1995). Nevertheless there is a difference concerning the shape parameter $b$ which in our case is not only defined for positive values but also for negative values. Such a case is disregarded by a classical 2-d Weibull-distribution.

Now we shortly summarize the mathematical descriptions of median, mode, mean, variance, and standard deviation parameters of $\mathcal{Z}$ for the case of a zero threshold. The calculations are described in detail in Appendix B for the general case of a non-zero threshold.

Median:

$$\nu = \left(\frac{\ln 2}{a}\right)^{1/b} \tag{12a}$$





Mode:

$$\eta = \left( \frac{b-1}{ab} \right)^{1/b} \quad \text{for } b > 1, b < 0 \; ; \quad \eta = 0 \text{ for } 0 < b \leq 1 \tag{12b}$$

Mean:

$$\mu = \frac{\Gamma\left(\frac{b+1}{b}\right)}{a^{\frac{1}{b}}} \tag{12c}$$

Variance and standard deviation:

$$\sigma^2 = \frac{\Gamma\left(\frac{b+2}{b}\right)}{a^{\frac{2}{b}}} - \mu^2 \; , \quad \sigma = \sqrt{\frac{\Gamma\left(\frac{b+2}{b}\right)}{a^{\frac{2}{b}}} - \mu^2} \tag{12d}$$

The expressions of mean and variance use the Gamma-function $\Gamma(t) = \int_0^\infty x^{t-1} e^{-x} dx$. Notice that the Gamma-function is defined for all real values of $t$ except $t = 0$ and all negative integer values of $t$. Note also that median, mode, mean, variance, and standard deviation parameters of $\mathcal{Z}$ coincide with those of an exponential distribution in the limit as $b$ equals one.

**5.2  Two computational methods to estimate the free parameters $a$ and $b$ of $\mathcal{Z}$ from a given data sample**

In this section we present two numerical methods to calculate the scale parameter $a$ and shape parameter $b$ describing the new $\mathcal{Z}$-distribution. First of all, since any measurement depends on a specific instrumental sensitivity, we have to introduce a threshold which we name $z_{th}$. The remaining data sample consists of N observations $z_i$ with $z_i > z_{th}$. Then we calculate the mean $\bar{m}$ and standard deviation $s$ of data $z_i$ using Eq. (3), and also the median value $\tilde{m}$ from data $z_i$.

Method (1): we investigate the corresponding theoretical moments from $\mathcal{Z}$. In Appendix B we derive the theoretical mean $\mu$ (Eq. B5) and median $\nu$ (Eq. B3) for the $\mathcal{Z}$-distribution with a threshold constraint. Taking the estimates of mean $\bar{m}$ and median $\tilde{m}$ from the sample as best proxies for the theoretical mean $\mu$ and median $\nu$ values of $\mathcal{Z}$, we get the following equations:

$$\nu = \left( \frac{\ln 2}{a} + z_{th}^b \right)^{1/b} \longrightarrow a = \frac{ln2}{\tilde{m}^b - z_{th}^b} \; , \tag{13a}$$

$$\mu = -\frac{A|b|\Gamma\left(\frac{b+1}{b}, az_{th}^b\right)}{ba^{1/b}} \longrightarrow 0 = \bar{m} \cdot ba^{1/b} + A|b|\Gamma\left(\frac{b+1}{b}, az_{th}^b\right) \; . \tag{13b}$$

Note that the use of a threshold constraint involves the introduction of a scaling factor $A = \exp(az_{th}^b)$ which is present in Eq. (13b). Inserting the algebraic term of $a$ (right side of Eq. (13a)) into the right side zero-equation (Eq. 13b) and using the threshold value of $z_{th}$ yields an equation only for $b$ which has to be computed iteratively. Once a numerical value of $b$ has been estimated with a sufficient accuracy, we insert this $b$ value into the upper right equation to get the numerical value for $a$.

We note that in classical statistics the method of moments determines $a$ and $b$ from the mean and variance equations. In principle this approach should be here possible too, but in practise the algebraic structure of the variance equation is too complicated, see Eq. (B6) in Appendix B. This means that the variance equation, if at all, is only iteratively solvable whereas the





use of the median equation offers an analytical transformation to $a$. Generally, we recommend to apply the proposed method using the mean and median equations. This straight-forward method is easy to program and produces reliable estimates of parameters $a$ and $b$.

Method (2): we also present a second method using a maximum likelihood approach, see Appendix C. The parameters are again calculated from two equations (Eq. C5) with

$$\frac{1}{a} = -z_{th}^{b} + \frac{\sum z_i^b}{N} \ , \quad 0 = \frac{1}{b} + a \cdot \ln z_{th} \cdot z_{th}^{b} + \frac{\sum \ln z_i}{N} - a \cdot \frac{\sum \ln z_i \cdot z_i^b}{N} \ . \tag{14}$$

Interestingly, the left equation includes a term $1/N \sum z_i^b$ which is the mean of the sample values weighted by power $b$ whereas the right equation includes the mean $1/N \sum \ln z_i$ of logarithmic data and $1/N \sum z_i^b \ln z_i$. This shows a similarity to the computation of regression points used in Figure 3. We insert $a$ into the right equation which yields a unique equation for $b$ which again can be solved iteratively. Once $b$ is fixed, the left equation allows to determine $a$. In the following we will test our lidar data samples with these two procedures and we will show that both methods produce almost identical results.

## 5.3 $\mathcal{Z}$-distributions applied to ALOMAR data

Applications of the $\mathcal{Z}$-distribution to ALOMAR data of maximum backscatter ($x$), ice mass density ($y$), ice particle radius ($r$) and ice number density ($n$) are shown in Figure 5. Note that thresholds have been computed from the regression functions (Eq. 5) described in Sect. 3.2 on the basis of $x_{th} = 3 \cdot 10^{-10} \mathrm{m}^{-1} \mathrm{sr}^{-1}$, and resulting $y_{th} = 22 \, \mathrm{mg/m}^3$, $r_{th} = 22.3 \, \mathrm{nm}$ and $n_{th} = 662 \, \mathrm{cm}^{-3}$ . The values of scale parameter $a$ and shape parameter $b$ have been calculated with the method of mean and median equations (method 1). Then the theoretical curves of $\mathcal{Z}$ and theoretical values of mean, median, mode and standard deviation have been calculated by inserting the values of $a$, $b$ and threshold $z_{th}$ into Eq. B1–B6. Obviously the pdf $\mathcal{Z}$ has sometimes no simple exponential shape which is the case for ice mass density, ice radius and ice number density. As we see in Figure 5 all $\mathcal{Z}$-pdf curves (in blue) match the original data histograms with a high accuracy. The relative error is in a range about 6-10 percent except that ice number density has a relative error of 15 percent. When we compare the mean, median, mode, and standard deviation derived from the theoretical distribution and corresponding estimates from data samples, we see a precise coincidence of mean and median values. Not surprising this is due to the fact that the parameters $a$ and $b$ have been computed by the mean and median method which guarantees the preservation of mean and median values. Nevertheless standard deviation and mode also show always a good agreement within the error range. A closer look to the maximum backscatter distribution shows that MBS data are almost perfectly exponentially distributed with $b = 0.931$ which is not too far away from $b = 1$ for an exact exponential pdf. As we had already shown, see Sect. 3.1.1, MBS data are very likely exponentially distributed, now the $\mathcal{Z}$–distribution analysis confirms this result. Hence we conclude that the commonly used exponential (g-function) analysis might be only a reasonable statistical method in case of analyzing MBS lidar data.

In contrast to MBS, the $\mathcal{Z}$-distribution of IMD shows a function that converges rapidly to zero for small IMD values. The distribution is described with $b = 1.355$ which significantly deviates from $b = 1$ for a precise exponential function. Note that the mode of the data sample at $40 \, \mathrm{mg/m}^3$ differs from the theoretical mode of $23 \, \mathrm{mg/m}^3$ because of a relatively large statistical noise in the data. But mean, median and standard deviation values agree almost perfectly. Similar to IMD, the ice radius





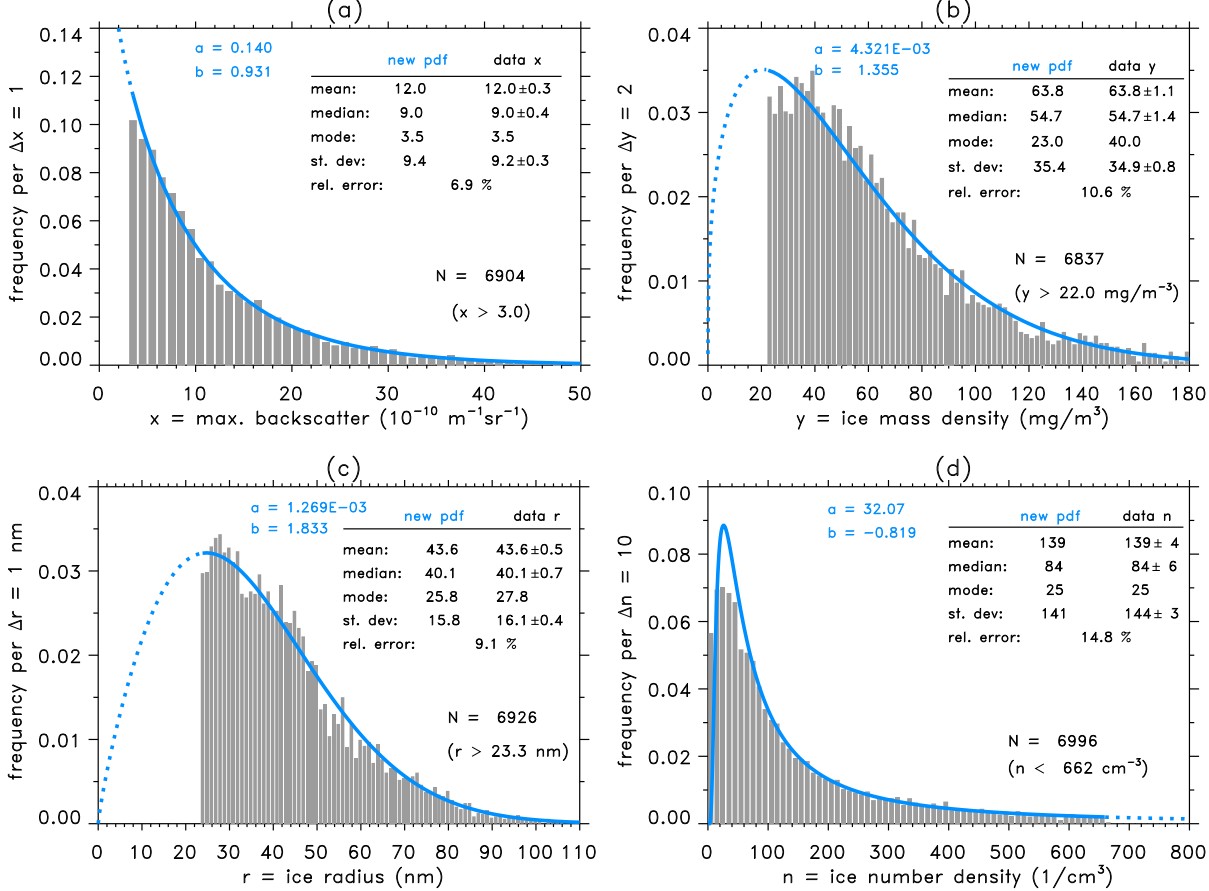

**Figure 5.** Frequency distributions and Z-function analysis of ALOMAR data. Parameter $a$ and $b$ have been estimated with the mean and median method. The relative error given in percent describes the quality of the Z-function fit. (a) Maximum backscatter data $x$. (b) Ice mass density $y$. (c) Ice particle radius $r$. (d) Ice number density $n$, see text for more details.

distribution indicates a significant non-exponential behavior with $b = 1.833$. The distribution converges to zero as the radius approaches zero. The curve is skewed to the right and has a maximum at $r = 25.8$ nm which differs only slightly from the mode of the data sample at $r = 27.8$ nm. Again mean, median and standard deviation values agree almost perfectly.

The sample of ice number density shows a completely different behavior with a slope parameter that is negative with $b = -0.819$. The physical meaning is that the parameter ice number density is negatively correlated with all other ice parameters. For example, large ice numbers $n$ correspond to small ice radii, IMD and MBS values. As a consequence this leads to a threshold of $n$ in the reverse direction, that is from large values to small values defined by $n < n_{th} = 662$ cm$^{-3}$. One can see this feature in the right tale of $\mathcal{Z}(n)$ plotted as a dashed curve, see Figure 5d. The reversal behavior is also present for small values of $n$. Small values of $n$ are measured for very bright PMC events with large MBS that have small occurrence rates. Therefore, the number of small ice particles has a relatively high uncertainty due to their low occurrence frequency, and it is this statistical





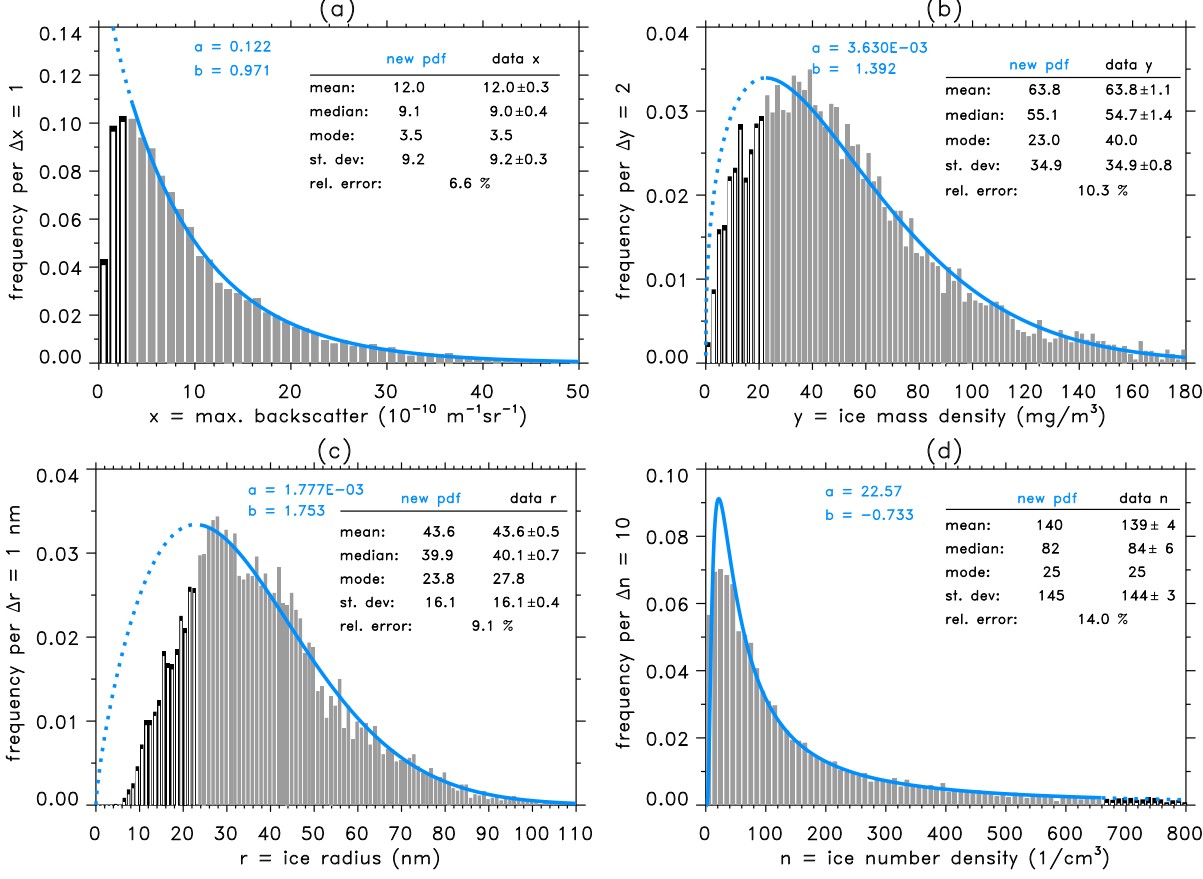

**Figure 6.** Same as Figure 5, but parameters $a$ and $b$ have been estimated from the maximum likelihood method. (a-c) Maximum backscatter, ice mass density and ice radius: gray bars indicate values larger than the threshold, black bars indicate values smaller than the threshold. (d) Ice number density: gray bars indicate values smaller than the threshold, black bars indicate values larger than the threshold.

error which produces some deviations from the fit curve to the data in the range of $n = 0 \text{-} 80 \, \text{cm}^{-3}$. We note that the numerical procedure computing the pair $(a, b)$ from the method of mean/median (Eq. 13a,b) has automatically detected the existence of a negative slope parameter $b$ without any a priori information.

Now we repeat the analysis using method 2. Figure 6 summarizes the $(a,b)$-values and statistical moments calculated from the method of maximum likelihood estimators. As can be seen the maximum likelihood approach computes almost identical results for all ice parameters. We have added in Figure 6a-c also the histogram bars (in black) for all data being smaller than the threshold. Have in mind that the calculation of theoretical distribution curves is based exclusively on data larger than the threshold. Hence, decreasing or increasing a threshold will change the specific values of $a$ and $b$. Figure 6d shows the ice number density distribution where we have added in the histogram (in black) all data being larger than the threshold. Again,





also the maximum likelihood method has automatically detected the existence of a negative slope parameter $b$ for the ice number density distribution.

## 6 Discussion

### 6.1 Construction of artificial data

In the derivation of the $\mathcal{Z}$-distribution we used the assumption that all ice parameters of maximum backscatter $x$=MBS, ice mass density $y$=IMD, ice particle radius $r$ and ice number density $n$ are connected with one another by the power law given in Eq. (5). In Sect. 5.3 we showed that the $\mathcal{Z}$-pdf describes with a high accuracy each distribution of these ice parameters which, in turns, means that indeed there exists at least an approximative power law between ice parameters. We discuss a suitable justification of this power law relation in more detail in Sect. 6.2. In the following we will show that the use of a $\mathcal{Z}$-distribution

allows to construct artificial unknown data samples of various ice parameters which approximate true data to a high degree. We think that such an application is one of the most beneficial outcome from the new $\mathcal{Z}$-distribution approach. We explain the numerical procedure by the help of an practical example.

We already showed a linear dependance in the logarithmic frame using linear regression (LR) for maximum backscatter and ice particle radius, see Figure 3b. Hence we can compute artificial ice radius proxies $\tilde{r}_i$, named as LR-proxy of true data $r_i$,

as a function of MBS-data $x_i$ from the regression power law function (Eq. 5) with $\tilde{r}_i = c x_i^d$ and with power law coefficients $c = 13.509$ and $d = 0.497$. Figure 7a shows a comparison between LR-proxy and original ice radius data where we test the identity of the two data samples. The correlation coefficient is the same as shown in Figure 3b with $R = 0.55$. Mean and median values of proxy and original data are almost identical, and a regression analysis shows a perfect identity ($c = 1.000$ and $d = 1.000$). Now we calculate the frequency histogram of LR-proxy ice radii, see Figure 7b, and compare the histogram with

the original $\mathcal{Z}$-distribution of ice radii already shown in Figure 5c. We find that the LR-proxy approximates the mean, median, mode and standard deviation values of the original $\mathcal{Z}$-distribution with an relative error of 9.5% comparable to the original error of 9.1%. We conclude that a linear regression analysis of logarithmic data offers a good opportunity to approximate data provided that a pair of data samples exists that allows the calculation of power law coefficients $c$ and $d$ from regression methods.

Now the $\mathcal{Z}$-distribution approach offers a more general possibility to derive artificial data samples without any knowledge of correlation and regression coefficients. Indeed we will show that results from the $\mathcal{Z}$-approach are very close to results from a regression analysis. Again, our goal is to approximate ice radius data from a given maximum backscatter data sample. But now we suppose that no data of ice particle radii $r$ exist, hence any correlation and regression analysis is not possible. First, we assume that a data sample of $x$ = MBS of number $N$ exists and also its $\mathcal{Z}$-distribution $\mathcal{Z}(x, a_x, b_x)$ with $a_x = 0.140$ and

$b_x = 0.931$ is well known, see Figure 5a. Secondly, we assume that we know a priori the form of the $\mathcal{Z}$-distribution $\mathcal{Z}(r, a_r, b_r)$ of ice radius $r$, e.g. with values of parameters $a_r$ and $b_r$ from Figure 5c ($a_r = 1.269 \cdot 10^{-3}$, $b_r = 1.833$). Have in mind that such information about scale and shape parameters of the ice radius distribution could be also provided from independent satellite measurements that are capable to measure ice particles radii, e.g. AIM-SOFIE.



**Figure 7.** (a) Proxy $p$ of ice radius versus original ice radius data without any threshold. The proxy has been derived from maximum backscatter data using the fit function that has been estimated by linear regression (LR-proxy) between original logarithmic MBS and ice radius data, see Figure 3b. (b) Frequency distribution of LR-proxy (gray and black histogram) with a threshold $r_{th} = 23.3$ nm. For comparison we also plot the original Z-pdf curve (blue) from the analysis of original ice radius data, see Figure 5c. The relative error describes the accuracy between LR-proxy data and original Z-function fit. (c) Same as a, but for Z-proxy data resulting from the Z-pdf analysis of MBS data, see text for more details. (d) Same as b with Z-proxy data.



Our new proxy method ($\mathcal{Z}$-proxy) requires the following transformations. We first transform the $x_i$-values ($i = 1, ..., N$) into the $z$-domain with $x_i \mapsto z_i$: $z = a_x x_i^{b_x}$ followed by a second transformation with $z_i \mapsto r_i$: $r_i = (z_i/a_r)^{1/b_r}$ resulting in

$$x_i \mapsto z_i \mapsto r_i : \quad r_i = \left[\frac{a_x \cdot x_i^{b_x}}{a_r}\right]^{1/b_r} = c \cdot x_i^d \quad \text{with} \quad c = \left(\frac{a_x}{a_r}\right)^{1/b_r} \quad , \quad d = \frac{b_x}{b_r} . \tag{15}$$

Note that the derivation of $c$ and $d$ in Eq. (15) is based on the same mathematical steps when we developed the $\mathcal{Z}$-distribution
from Eq. (8) to Eq. (9). Inserting the $a_x$, $b_x$, $a_r$ and $b_r$ values into Eq. (15) determines the power law coefficients for $\mathcal{Z}$-proxy $\tilde{r}$ with $c = 12.994$ and $d = 0.508$. These values do not exactly coincide with $c$ and $d$ values obtained from the regression method, see above, but the identity test between $\mathcal{Z}$-proxies and true ice radii shows a very good coincidence, see Figure 7c. Again, mean and median values of proxy and original data are practically identical, and a regression analysis shows almost a perfect identity ($c = 1.095$ and $d = 0.980$). Finally we calculate a frequency histogram of $\mathcal{Z}$-proxies, see Figure 7d, and find a good agreement
between proxies and true pdf. Mean, median, and standard deviations of $\mathcal{Z}$-proxy data correspond perfectly to original ice radius data, and the relative error has now even decreased to 9.2%.

We summarize that we present a new method in order to construct artificial data samples provided $\mathcal{Z}$-descriptions of these data sets exist. By means of a consecutive arranging of ice parameters starting at one given data sample this method allows to construct any artificial data sample within $(x, y, r, n)$. This method can be also applied to other data sets, e.g. ice parame-
ter measurements from satellite observations. For example, a data sample of ice water content (IWC) obtained from satellite measurements might be analyzed in terms of a $\mathcal{Z}$-distribution estimating the scale and shape parameters $a$ and $b$ of the IWC distribution. This would allow to establish a connection of satellite IWC data to lidar data samples $(x, y, r, n)$ through Eq. (15), hence the satellite IWC data could be transferred to lidar maximum backscatter, ice mass density, ice particle radius and ice number density. Vice versa the knowledge of an satellite IWC $\mathcal{Z}$-distribution would allow to transform lidar observations
into IWC proxies and compare these with the original IWC observed by the satellite. We think that our proposed transformation method could be very helpful to connect different ice parameter data from different instruments, either from satellite observations or ground based measurements. We also think that this new approach might be important in trend analysis of NLC/PMC.

In the next section we will discuss the power law assumption (Eq. 5) and the physical meaning of the shape parameter $b$
which might be introduced as a new trend variable in the analysis of NLC/PMC long-term changes.

## 6.2   Discussion of the power law assumption between PMC parameters

In this section we discuss some theoretical aspects of the power law dependence on ice parameters in order to validate the justification of Eq. (5). We use again the assumption as already discussed in Sect. 2 that at the altitude of maximum brightness (MBS) and ice mass density (IMD) there exists in the real atmospheric background an ice particle distribution that is perfectly
Gaussian ($\mathcal{N}_i$) distributed as

$$\mathcal{N}_i(\tau) = \frac{1}{\sigma_i \sqrt{2\pi}} \exp(-\frac{1}{2}\left(\frac{\tau - r_i}{\sigma_i}\right)^2) .$$





We also assume that the geometric shapes of these ice particles are spheres with ice radii $\tau$ with mean radius $r_i$ and variance $\sigma_i^2$. Again index $i = 1, ..., N$ relates to the $i$-th measurement in a given data sample of number $N$. $\mathcal{N}_i$ is normalized to $\int_0^\infty \mathcal{N}_i(\tau) d\tau = 1$. When we assume an ice number density of $n_i$ particles per cm³ we get the expression $\int_0^\infty n_i \cdot \mathcal{N}_i(\tau) d\tau = n_i$. Note that mean ice radii $r_i$ and ice number densities $n_i$ are elements of our lidar data climatology which we have introduced

in Sect. 2.

Furthermore, we assume from the analysis of lidar observations (3-color measurements) that the relation of mean radius and variance is according to $\sigma_i = 0.4 r_i$ for $r_i > 37.5$ nm and $\sigma_i = 15$ nm for $r_i \geq 37.5$ nm (*Baumgarten et al.*, 2010). This assumption has been also applied in the analysis of AIM/SOFIE-CIPS PMC satellite data (*Lumpe et al.*, 2013; *Hervig and Stevens*, 2014). In order to simplify calculations we apply this relation $\sigma_i = 0.4 r_i$ also for $r_i$ larger than 37.5 nm. We now

investigate the question which backscatter lidar and ice mass signal result from such an ice distribution?

We compute the mass of an spherical ice particle with radius $\tau$ as $4/3 \pi \rho_{ice} \tau^3$ with density of ice $\rho_{ice} = 932$ kg/m³. The backscatter signal from a single ice particle is calculated as $a_L \tau^{5.8}$ with the lidar constant $a_L = 1.5 \cdot 10^{-11}$ m². Then the maximum backscatter $x_i$ and ice mass density $y_i$ are estimated by an integration of the radius distribution from zero to infinity as

$$x_i = a_L n \int_0^\infty \tau^{5.8} \mathcal{N}_i(\tau) d\tau = a_L n \int_0^\infty \tau^{5.8} \frac{1}{0.4 r_i \sqrt{2\pi}} \exp\left(-\frac{1}{2}\left(\frac{\tau - r_i}{0.4 r_i}\right)^2\right) d\tau$$

$$y_i = \frac{4}{3} \pi \rho_{ice} n \int_0^\infty \tau^3 \mathcal{N}_i(\tau) d\tau = \frac{4}{3} \pi \rho_{ice} n \int_0^\infty \tau^3 \frac{1}{0.4 r_i \sqrt{2\pi}} \exp\left(-\frac{1}{2}\left(\frac{\tau - r_i}{0.4 r_i}\right)^2\right) d\tau$$

assuming a constant number density $n$ of ice particles. Only the integral of $y_i$ is analytically computable with a solution in which the error function defined by the integral $erf(x) = 2/\sqrt{\pi} \int_0^x \exp(-t^2) dt$ is part of the solution:

$$x_i = \frac{4}{3} \pi \rho_{ice} n \left[\frac{37}{50} r_i^3 erf\left(\frac{5(\tau - r_i)}{\sqrt{8} r_i}\right) - \frac{\sqrt{2}}{125 \sqrt{\pi}} r_i \left(25\tau^2 + 25 r_i \tau + 33 r_i^2\right) \exp\left(\frac{25\tau}{4 r_i} - \frac{25(\tau^2 + r_i^2)}{8 r_i^2}\right)\right]_0^\infty .$$

The integral for $x_i$ includes the term $\tau^{5.8}$ that arises from Mie-scatter theory for light scattering of a wavelength of 532 nm (ALOMAR RMR-lidar) at spheres in a range of radii with 1–100 nm. The exponential value of 5.8 approximates exact Mie-scatter calculations with an relative error less than 0.5 % in this radii range. Unfortunately, the integral can only be solved analytically if the exponent is an integer number as 5 or 6, respectively. Nevertheless, we are able to solve this integral by means of numerical methods with the specific exponent of 5.8. In a next step, we construct analytical approximations $f_x(r_i)$

and $f_y(r_i)$ for both integral solutions using a typical value of $n = 200$ cm⁻³ with

$$f_x(r_i) = x_i = a_1 a_L n r_i^{5.8} , \quad f_y(r_i) = y_i = a_2 \frac{4}{3} \pi \rho_{ice} n r_i^3 .$$

The linear constants $a_1$ and $a_2$ with values $a_1 = 4.20$ and $a_2 = 1.47$ are optimal dimensionless parameters. $f_y$ approximates the analytical solution of $y_i$ with a relative error less than 0.7 percent in the range $r_i = [0 \text{ nm}; 45 \text{ nm}]$, and less than 1.2 percent in the range $r_i = [45 \text{ nm}; 70 \text{ nm}]$. A precise solution of $x_i$ resulting from numerical methods of integration is approximated by $f_x$

with a relative error less than 0.3 percent in the range $r_i = [0 \text{ nm}; 45 \text{ nm}]$, then the relative error increases linearly to a maximum





error of 5 percent at $r_i = 70$ nm. We find that the solutions $f_x$ and $f_y$ approximate the general power law condition $p = cq^d$ (Eq. 5) inside a small error range. Hence these analytical examples show that MBS is a function of ice radius proportional to $\sim r^{5.8}$ ($d = 5.8$), the same is also true for IMD ($\sim r^3$, $d = 3$). It also follows that MBS and IMD are consequently connected through a power law condition with MBS $\sim$ IMD$^{5.8/3}$ ($d = 5.8/3 = 1.93$).

But the new form of the $\mathcal{Z}$-distribution technique opens up whole new perspectives for the validation of the analytical examples based on the ALOMAR lidar data samples. We transform the z-distribution of IMD into the MBS domain using Eq. (15) with $y_i \mapsto z_i$: $z = a_y y_i^{b_y}$ and $z_i \mapsto x_i$: $x_i = (z_i/a_x)^{1/b_x}$ that gives

$$x = \left[\frac{a_y \cdot y^{b_y}}{a_x}\right]^{1/b_x} = c \cdot y^d \quad \text{with} \quad c = \left(\frac{a_y}{a_x}\right)^{1/b_x} \quad , \quad d = \frac{b_y}{b_x} \ . \tag{16}$$

We insert into Eq. (16) the values of $a_x = 0.140$, $b_x = 0.931$, $a_y = 4.321 \cdot 10^{-3}$ and $b_y = 1.355$ from Figure 5a,b and get $c = 0.023$
and $d = 1.46$. Obviously, the power constant ($d = 1.46$) derived from the shape parameters $b_x$ and $b_y$ of the $\mathcal{Z}$-distribution analysis of real ALOMAR IBS and IMD data is sufficiently different from the power estimate ($d = 1.93$) belonging to the analytical example that necessitates various assumptions, e.g Gaussian distributed ice particles at the height of maximum backscatter, constant ice particle number or spherical shape of ice particles. Hence, we conclude that the determination of shape parameters $b$ from a $\mathcal{Z}$-distribution analysis therefore provides a qualitative indication of the actual microphysical state
that controls real ice formation processes. This leads to the idea that as a future task long-term changes in PMC formation might be characterized by potential long-term changes in $b$ that indicate long-term changes of atmospheric background conditions and microphysical ice constraints of ice formation.

## 7   Summary and conclusions

In this study we present a new method to describe statistical probability density distributions (pdfs) for different ice parameters
of PMC/NLC. We analyze a climatology of ice seasons from 2002 until 2016 as measured by the ALOMAR lidar. From this data set we derive ice cloud parameters of maximum backscatter, ice mass density, ice radius and ice number density whose occurrence frequencies are investigated with respect to exponential distributions. We show that only maximum backscatter follows an exponential distribution whereas ice mass density, ice radius and ice number density frequencies fail to fit satisfactorily to an exponential distribution. The reason for these deviations from exponential behavior is based on the fact that these
ice parameters are not linearly dependent on each other.

     We introduce a new probability density distribution ($\mathcal{Z}$-function, see Eq. 9) that assumes instead a general power law relation among ice parameters, see Eq. (5). The new $\mathcal{Z}$-distribution is described by two free constants with scale parameter a and the shape parameter b. We point out that the new distribution is closely related to a more general Weibull-distribution. The new distribution has been applied to maximum backscatter, ice mass density, ice radius and ice number density data from the
ALOMAR data set. As a result all data distributions are described with a high accuracy by $\mathcal{Z}$. We discuss that the exponential distribution (g-function) is a special case of the more general $\mathcal{Z}$-function with shape parameter $b = 1$. We present two numerical




stable methods (method of mean and median, method of maximum likeliness) that allow to derive the values of free constants a and b describing the actual $\mathcal{Z}$-function shape for a given data sample.

Perhaps the most important application of the new method is the possibility to construct unknown data sets for different ice parameters which approximate true data to a high degree. We show in Sect. 6.1 that a linear regression analysis in a logarithmic

data frame offers a good opportunity to approximate data provided that a pair of data samples exists that allows the calculation of power law coefficients $c$ and $d$ from regression methods. The $\mathcal{Z}$-distribution approach offers a more general possibility to derive artificial data samples without any knowledge of correlation and regression coefficients. This allows the connection of different observational PMC distributions of lidar, and satellite data, and also with distributions resulting from ice model studies. In particular, the statistical distributions of different measured ice parameter can be compared with each other on the

basis of a common assessment that again should be helpful in combining trend analysis of PMC/NLC long term time series from different observational data sets

## Appendix A: Properties of the exponential distribution (g-function)

When considering a threshold ($x_{th}$) the exponential pdf $\mathcal{E}(x)$ is normalized according to $A \int_{x_{th}}^{\infty} \mathcal{E}(x)dx = 1$ with a scaling factor $A = \exp(\alpha x_{th})$. It follows that the mean $\mu$ is then given by

$$\mu = \int_{x_{th}}^{\infty} x \cdot A\mathcal{E}(x) \, dx' = A \int_{x_{th}}^{\infty} x \cdot \alpha e^{-\alpha x} \, dx' = -A \frac{(\alpha x + 1) e^{-\alpha x}}{\alpha} + C$$

$$= 0 - (-) e^{\alpha x_{th}} \frac{(\alpha x_{th} + 1) e^{-\alpha x_{th}}}{\alpha} \,.$$

This yields for the mean

$$\mu = x_{th} + 1/\alpha. \tag{A1}$$

The median $\nu$ denotes the boundary of separating the higher half from the lower half of the distribution with

$$0.5 = \int_{\nu}^{\infty} A \cdot \mathcal{E}(x) \, dx' = A \int_{\nu}^{\infty} \alpha \exp(-\alpha x) \, dx' = -A e^{-\alpha x} + C$$

$$= 0 - (-) e^{\alpha x_{th}} \cdot e^{-\alpha \nu} \,.$$

The equation is solved for the median with

$$\nu = x_{th} + ln(2)/\alpha \,. \tag{A2}$$

The mode is the value $\eta$ at which $A\mathcal{E}(x)$ takes its maximum value

$$\eta = x_{th} \,. \tag{A3}$$





The variance $\sigma^2$ of an exponential distribution considering a threshold is calculated with

$$
\begin{aligned}
\sigma^2 &= \int_{x_{th}}^{\infty} (x-\mu)^2 \cdot A\mathcal{E}(x)\, dx' = A\int_{x_{th}}^{\infty} (x-\mu)^2 \cdot \alpha e^{-\alpha x}\, dx' \\
&= -\frac{A\left(\alpha\left(x-\mu\right)\left(\alpha\left(x-\mu\right)+2\right)+2\right)e^{-\alpha x}}{\alpha^2} + C \\
&= 0 - (-)e^{\alpha x_{th}} \frac{\left(\alpha\left(x_{th}-\mu\right)\left(\alpha\left(x_{th}-\mu\right)+2\right)+2\right)e^{-\alpha x_{th}}}{\alpha^2} .
\end{aligned}
$$

Inserting $\mu = x_{th} + 1/\alpha$ simplifies the algebraic expression and shows that the variance $\sigma^2$ (standard deviation $\sigma$) is independently from a given threshold:

$$
\sigma^2 = 1/\alpha^2 . \tag{A4}
$$

## Appendix B: Properties of $\mathcal{Z}$-distribution

In the following all quantities take into account a threshold $z_{th}$. We introduce a scaling factor $A = \exp(az_{th}^b)$. Setting the threshold to zero means a scaling factor $A = 1$ and gives the regular expressions for cumulative pdf, median, mode, mean, and variance, see Eq. (12a–d).

Probability density function $\mathcal{Z}(z \geq z_{th}, a > 0, b \neq 0)$:

$$
\mathcal{Z}(z) = A \cdot a|b|z^{b-1}e^{-az^b} , \quad 1 = \int_{z_{th}}^{\infty} e^{az_{th}^b} \cdot a|b|z^{b-1}e^{-az^b}\, dz . \tag{B1}
$$

Cumulative form of $\mathcal{Z}_{cum}$:

$$
\mathcal{Z}_{cum}(z) = A\int_{z}^{\infty} a|b|z^{b-1}e^{-az^b}\, dz = -Ae^{-az^b} + C = A \cdot e^{-az^b} , \quad z \geq z_{th} . \tag{B2}
$$

Median $\nu$:

$$
0.5 = \int_{\nu}^{\infty} \mathcal{Z}\, dz = \int_{\nu}^{\infty} Aa|b|z^{b-1}e^{-az^b}\, dz = -Ae^{-a\nu^b} + C = Ae^{-a\nu^b}
$$

$$
\longrightarrow \nu = \left(\frac{\ln(2A)}{a}\right)^{1/b} = \left(\frac{\ln 2 + \ln(e^{az_{th}^b})}{a}\right)^{1/b} = \left(\frac{\ln 2}{a} + z_{th}^b\right)^{1/b} . \tag{B3}
$$

Mode $\eta$:

$$
\frac{\partial \mathcal{Z}}{\partial z} = 0 = -A \cdot a|b|z^{b-2}\left(abz^b - b + 1\right)e^{-az^b} \longrightarrow \eta = \left(\frac{b-1}{ab}\right)^{1/b} \quad (b > 1, b < 0) . \tag{B4}
$$



Mean $\mu$:

$$\mu = \int_{z_{th}}^{\infty} z \mathcal{Z} dz = A \int_{z_{th}}^{\infty} a|b|z^b e^{-az^b} dz = -\frac{A|b|\Gamma\left(\frac{b+1}{b}, az_{th}^b\right)}{ba^{\frac{1}{b}}} . \tag{B5}$$

Details of calculation:

Substitute: $\quad u = a^{\frac{b+1}{b}} z^{b+1} \longrightarrow dz = \frac{1}{(b+1)a^{\frac{b+1}{b}} z^b} du$

$$\int Aa|b|z^b e^{-az^b} dz = \frac{Aa|b|}{ba^{\frac{1}{b}+1} + a^{\frac{1}{b}+1}} \cdot \int e^{-u^{\frac{b}{b+1}}} du$$

We solve: $\quad \int e^{-u^{\frac{b}{b+1}}} du = -\frac{(b+1)\Gamma(\frac{b+1}{b}, u^{\frac{b+1}{b}})}{b}$

Inserting: $\quad \frac{Aa|b|}{ba^{\frac{1}{b}+1} + a^{\frac{1}{b}+1}} \cdot \int e^{-u^{\frac{b}{b+1}}} du = -\frac{Aa|b|(b+1)\Gamma(\frac{b+1}{b}, u^{\frac{b+1}{b}})}{b(ba^{\frac{1}{b}+1} + a^{\frac{1}{b}+1})}$

Resubstitute: $\quad = -\frac{Aa|b|(b+1)\Gamma(\frac{b+1}{b}, az^b)}{b(ba^{\frac{1}{b}+1} + a^{\frac{1}{b}+1})} = -\frac{A|b|\Gamma\left(\frac{b+1}{b}, az^b\right)}{ba^{\frac{1}{b}}} + C .$

Here we use the Gamma-function $\Gamma(a) = \int_0^{\infty} t^{a-1} e^{-t} dt$ and the incomplete Gamma-function $\Gamma(a,x) = \int_x^{\infty} t^{a-1} e^{-t} dt$. Notice that the Gamma-function is defined for all real values of $a$ except $a = 0$ and all negative integer values of $a$. The same applies to $\Gamma(a,x)$ with $x \geq 0$.

Variance $\sigma^2$:

$$\sigma^2 = \int_{z_{th}}^{\infty} (z-\mu)^2 \mathcal{Z} dz = A \int_{z_{th}}^{\infty} ab(z-\mu)^2 z^{b-1} e^{-az^b} dz$$

$$= -A\mu^2 e^{-az^b} - \frac{A\left[\Gamma\left(\frac{b+2}{b}, az^b\right) - 2a^{\frac{1}{b}}\mu\Gamma\left(\frac{b+1}{b}, az^b\right)\right]}{a^{\frac{2}{b}}} + C$$

$$= \frac{A\left[\Gamma\left(\frac{b+2}{b}, az_{th}^b\right) - 2a^{\frac{1}{b}}\mu\Gamma\left(\frac{b+1}{b}, az_{th}^b\right)\right]}{a^{\frac{2}{b}}} - \mu^2 ,$$

for $A = 1, z_{th} = 0: \quad \sigma^2 = \frac{\Gamma\left(\frac{b+2}{b}\right)}{a^{\frac{2}{b}}} - \mu^2 = \frac{\Gamma\left(\frac{b+2}{b}\right) - \Gamma^2\left(\frac{b+1}{b}\right)}{a^{\frac{2}{b}}} = \frac{2\Gamma\left(\frac{2}{b}\right) - \frac{1}{b}\Gamma^2\left(\frac{1}{b}\right)}{ba^{\frac{2}{b}}} . \tag{B6}$

## Appendix C: Estimation of parameters (a,b) using the maximum log-likelihood method

For a single observation, the likelihood function $l$ of $Z$ is calculated from Eq. (B1). Given a sample of N observations with threshold $z_{th}$, the likelihood function $l(a,b) = \prod_{i=1}^{N} \mathcal{Z}(z_i)$ is

$$l(a,b) = e^{Naz_{th}^b} (a|b|)^N \prod_{i=1}^{N} z_i^{b-1} e^{-az_i^b} . \tag{C1}$$




Taking the logarithm of $l$ yields the log-likelihood function $L(a,b) = \ln(l) = \sum_{i=1}^{N} \ln(\mathcal{Z}(z_i))$

$$L(a,b) = Naz_{th}^b + N(\ln a + \ln|b|) + (b-1)\sum_{i=1}^{N}\ln(z_i) - a\sum_{i=1}^{N} z_i^b . \tag{C2}$$

The derivative with respect to parameter $a$ is

$$\frac{\partial L(a,b)}{\partial a} = N \cdot z_{th}^b + \frac{N}{a} - \sum_{i=1}^{N} z_i^b, \tag{C3}$$

and for parameter $b$

$$\frac{\partial L(a,b)}{\partial b} = N \cdot a \cdot \ln z_{th} \cdot z_{th}^b + \frac{N}{b} + \sum_{i=1}^{N}\ln z_i - a\sum_{i=1}^{N} z_i^b \ln z_i . \tag{C4}$$

Setting each of the derivatives equal to zero yields for $a$ and $b$

$$\frac{1}{a} = -z_{th}^b + \frac{\sum z_i^b}{N} \quad , \quad 0 = \frac{1}{b} + a \cdot \ln z_{th} \cdot z_{th}^b + \frac{\sum \ln z_i}{N} - a \cdot \frac{\sum \ln z_i \cdot z_i^b}{N} . \tag{C5}$$

These are the maximum-likelihood estimators for scale parameter $a$ and shape parameter $b$.

*Competing interests.* The authors declare that they have no conflict of interest.

*Acknowledgements.* We appreciate the financial support from the German BMBF for the ROMIC/TIMA project. We thank Gary E. Thomas for very helpful and stimulus contributions, discussions and reviewing.




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
