# Peer review of "A new Description of Probability Density Distributions of Polar Mesospheric Clouds (PMC)"

_Atmospheric Chemistry and Physics, 2018_

## Referee Comment (RC1) · Anonymous Referee #1 · 13 Aug 2018

GENERAL SUMMARY AND COMMENTS

This paper presents a new analysis of the statistical behavior of various polar mesospheric cloud (PMC) properties, using observational data from the ALOMAR lidar in Norway. The authors show that the commonly used g-distribution, which prescribes an exponential dependence for the cumulative probability of a given parameter value, is not adequate for the full range of all PMC quantities determined from lidar data. A revised probability density function called the Z-distribution is developed that has two free parameters (scale and shape), and simplifies to the g-distribution when shape = 1. The new function provides a better representation of quantities such as ice mass density and ice radius, and is also shown to enable the creation of "artificial" data from one parameter into a second parameter given Z-distributions of both quantities.

[Figure]

This work is a valuable extension of the original g-distribution concept. However, it is not clear that the proposed applicability to long-term trend studies is justified. Some suggestions and comments related to specific items are provided below.

SPECIFIC COMMENTS

1. p. 2, line 1: Since the monograph cited here may not be easily accessible for many readers, I suggest reproducing the relevant figure (with permission) to help introduce the basic concept of the g-distribution.

2. p. 3, line 8: This choice obscures the inter-annual variability in PMC behavior that has been demonstrated in many previous studies (e.g. Rong et al. [2014], DeLand and Thomas [2015], Fiedler et al. [2017]), which can represent a factor of two variation for the slope of the g-distribution at the latitude of ALOMAR. How does this averaging affect the applicability of the results derived later to any specific PMC season?

3. p. 3, line 10: Is the three-color mode of operation used less often? Other papers discussing ALOMAR lidar measurements talk about 15-minute binning (e.g. Fiedler et al. [2017]), so I would actually expect many more individual profiles to be available from 15 years of data.

4. p. 3, line 31: I'm not sure I understand how the standard deviation can be equal to the mean with a threshold of zero. Statistically, one sigma should not encompass all data smaller than the mean, but with the definition given in line 29, you are not allowing for negative values of x. So if (mean – st.dev.) = 0, where does (mean – 2*sigma) fall?

5. p. 5, line 6: I'm not sure that the term "obviously" is appropriate. The fit line in Figure 2a does go through the data, but the fluctuations between y = 40-90 look comparable in magnitude to those between x = 20-40 in Figure 1a. The rolloff at y < 40 in Figure 2a is more significant to me, and it suggests that using a higher threshold (e.g. 40) would yield a satisfactory fit.

6. p. 5, lines 17-18: Regarding "larger discrepancies", see comment #5.

[Figure]

7. p. 7, lines 3-4: MBS is a first-level measured quantity, whereas IMD, R, and n are derived based on various assumptions. Does this "failure" say something about the functional forms used to create the latter group of products?

8. p. 8, lines 2-3: This figure uses data well below the previously defined fit threshold for both MBS and IMD. Does the result change if MBS > 3, IMD > 20 are required as specified for Figures 1 and 2?. What about IMD > 40, as suggested in comment #5?

9. p. 13, lines 15-16: The first two derived threshold values are close to those given in Section 3.1.2. Is the third value a maximum?

10. p. 13, line 33: This statement seems to connect back to lines 23-24 on this page. Isn't it circular reasoning to say that they agree?

11. p. 14, lines 5-6: This statement is physically plausible for radius. It seems reasonable for IMD which is proportional to $r^3$. Not sure about MBS, because it seems like large density could overcome the dependence on r (but is this true if MBS is proportional to $r^6$?).

12. p. 18, lines 17-19: I'm still not convinced that the parameters derived from a multi-season collection of data are valid to use for this type of "synthetic" data calculation with a smaller subset of original IWC data, based on the previous comments about interannual variations.

13. p. 20, lines 10-13: Is this statement saying that the uncertainty in the retrieval assumptions is large enough to justify the difference in b? What level of agreement would be needed for confidence?

14. p. 20, lines 15-17: This goal would require quantitative answers to comment #13 in order to be able to identify such changes. It also goes back to comment #2 regarding the question of how these fits behave with different individual years of data, and discussing how much noise increases with the reduction in the number of samples.

TYPOGRAPHICAL ERRORS

p. 5, lines 14-15: "unequal" could be "not equal to".

p. 5, line 15: "unequal" could be "not equal to".

p. 8, line 26: "allows to" should be "allows us to".

p. 9, line 9: "particulary" should be "particularly".

p. 14, line 8: "tale" should be "tail".

p. 15, line 7: "Have in mind" could be "Please keep in mind".

p. 16, line 11: "outcome" should be "outcomes".

p. 24, line 12: "stimulus" could be "stimulating".

---

## Referee Comment (RC2) · Anonymous Referee #2 · 23 Sep 2018

General comments:

This is an interesting and generally well written article dealing with probability density functions of various noctilucent cloud (NLC) parameters such as particle radius, cloud backscatter, ice particle density and ice mass density. While the backscatter is found to follow an exponential distribution, this is not the case for the other parameters considered. The NLC parameter database employed is based on the well-known Alomar LIDAR dataset. I do not have major objections against the publication of this article but ask the authors to consider the comments listed below. In addition, I have the following general comment: The LIDAR backscatter measurements, like all other optical measurements, are quite insensitive to particles with radii below a certain threshold. This is related to the finding that radii below about 20 nm are very infrequent in the data

[Figure]

set, despite the fact that there are typically many more small particles than large particles. I think this aspect should be discussed in the paper, because it also (qualitatively) explains some of the differences between the PDFs of the different parameters.

Specific comments:

Page 1, line 1 and line 15: "of Polar Mesospheric Clouds (PMC) and noctilucent clouds (NLC)."

This sounds like the two are different clouds. I suggest changing this sentence.

Page 1, line 5: "previously statistical methods" -> "previous statistical methods" or "previously used statistical methods"

Page 1, line 6: "probability statistic"

Does "statistic" exist?

Page 1, line 12: "that facilitate" -> "that facilitates", because "facilitates" refers to "assessment", right?

Page 2, line 3: "many .. analysis" -> "many .. analyses"

Page 2, line 8: "analysis have used" -> "analyses have used"

Page 2, line 17 and line 18: "statistic" ?

Page 2, line 31: "From each backscatter height profile we estimate a maximum backscatter (MBS) signal which corresponds to mean height of maximum brightness"

I don't fully understand this sentence. It mixes "signal" and "height" in a way, which makes it difficult to understand. Can you clarify, please?

Page 3, line 15: "exponential distributed" -> "exponentially distributed"

Page 3, line 30: "mode" is not a really frequently used term and I suggest briefly explaining it. It is explained on the next page and I suggest moving the explanation

here.

Page 4, line 19: "in a semi-logarithm scale". I suggest replacing this by "in a semi-logarithmic diagram" (a scale can be linear or logarithmic, but not semi-logarithmic)

Page 4, line 22: "Consequently, the relative error is rather small"

Please explain briefly how this relative error is determined.

Page 5, line 5: " . . . as expected"

It's not entirely clear, what you consider to be expected. Do you expect that these other parameters also follow an exponential distribution or do you not? Please clarify.

Page 5, line 6: "in a semi-logarithmic scale" -> "in a semi-logarithmic diagram"

Page 5, line 9: "significant smaller" -> "significantly smaller"

Page 6, Caption Fig. 2, line 2: "least square fit" -> "least squares fit"

Page 7, Fig. 3: I suggestion mentioning in the Figure caption what the dashed lines are.

Page 7, line 7: "Linearity between maximum backscatter (MBS) and ice mass density (IMD), ice radius r and ice number density n data is a necessary and sufficient condition that also IMD, r and n data samples are exponentially distributed"

I'm not sure you would really expect that MBS scales linearly with, e.g. radius. The intensity of the backscattered radiation does certainly not scale linearly with particle radius, right? Why should the maximum backscatter depend linearly on radius? If there are other indications etc. for that, please discuss. Considering that you use a power law to describe the relationship between two parameters, you don't really assume linearity, right? I think the term "linearity" should be replaced and then all is fine.

Page 8, line 1: ". . . also relate to the half width of the angle"

Can you mention how the regression points "relate to" the half width of this angle? To

me this is not obvious, but perhaps I'm missing something.

Page 8, line 5: "criteria" -> "criterion"

Page 8, line 6: "which is far away from unity"

This is not surprising at all, because the backscatter does not scale linearly with radius. But perhaps this is discussed below.

Page 9, last line: Something is missing in this equation. "C" is not defined and it is neither required here. The integral can be explicitly evaluated, but I find that $b>1$ is a requirement for the integral being 1. Please check.

Page 10, line 10: Meaning of "Only, " at beginning of sentence not clear, at least to me. Without the comma it would make sense.

Page 12, line 26: "should be here possible too" -> "should be possible here too"

Page 13, line 15: Suggest replacing ", and resulting" by ", resulting in "

Page 16, line 8: "in turns" -> "in turn"

Page 16, line 12: "of an practical example" -> "of a practical example"

Page 16, line 25: "Now the Z-distribution approach offers a more general possibility to derive artificial data samples without any knowledge of correlation and regression coefficients."

I'm not sure I fully agree with this statement, because information on the power law relationship between the two quantities is required, right? The statement suggests (or may suggest) that no prior information on the relationship between the two quantities is needed, which is certainly not the case, because you assume that $a\_r$ and $b\_r$ are known.

Page 18, line 19: "of an satellite" -> "of a satellite"

Page 19, line 7: I think "for $r\_i > 37.5$ nm" should be "for $r\_i < 37.5$ nm"

Interactive
comment

Page 19, line 10: "signal result" -> "signals result"

Page 19, line 11: "of an spherical ice particle" -> "of a spherical ice particle"

Page 19, line 11/12: you assume a fixed relationship between LIDAR backscatter signal and particle radius. In reality, the power will decrease with increasing particle radius. For your estimation this certainly does not have to be considered, but it's perhaps worth mentioning.

Page 21, last line: "We present two numericalLY stable .."

Page 21, line 6: "The Z-distribution approach offers a more general possibility to derive artificial data samples without any knowledge of correlation and regression coefficients"

OK, but the approach requires a priori knowledge on the Z-distribution parameters, right?

---

## Author Comment (AC1) · 8 Nov 2018

**Response to Anonymous Referee #1**

Journal: ACP
Title: A new Description of Probability Density Distributions of Polar Mesospheric Clouds (PMC)
Author(s): Uwe Berger et al.
MS No.: acp-2018-642
MS Type: Research article
Special Issue: Layered phenomena in the mesopause region (ACP/AMT inter-journal SI)

*Introductory remarks:*
*We greatly appreciate the comments from the reviewer. We have taken his/her suggestions for improvements into account when preparing the revised version of the manuscript. In the following we respond to the reviewer's comments point by point.*
*We have marked the changes in the tracked version of the manuscript. Author responses are in italics. Line numbers refer to the first paper version. In the new tracked version deleted sequences are marked* red. *New text is marked in* blue.
*We want to thank the reviewer for the detailed review with many useful ideas and suggestions which, we think, have significantly increased the quality of the manuscript.*

**GENERAL SUMMARY AND COMMENTS:**

This paper presents a new analysis of the statistical behavior of various polar mesospheric cloud (PMC) properties, using observational data from the ALOMAR lidar in Norway. The authors show that the commonly used g-distribution, which prescribes an exponential dependence for the cumulative probability of a given parameter value, is not adequate for the full range of all PMC quantities determined from lidar data. A revised probability density function called the Z-distribution is developed that has two free parameters (scale and shape), and simplifies to the g-distribution when shape =1. The new function provides a better representation of quantities such as ice mass density and ice radius, and is also shown to enable the creation of "artificial" data from one parameter into a second parameter given Z-distributions of both quantities.

This work is a valuable extension of the original g-distribution concept. However, it is not clear that the proposed applicability to long-term trend studies is justified. Some suggestions and comments related to specific items are provided below.

SPECIFIC COMMENTS:

1. p. 2, line 1: Since the monograph cited here may not be easily accessible for many readers, I suggest reproducing the relevant figure (with permission) to help introduce the basic concept of the g-distribution.
*We agree but given the number of figures already in the manuscript we see no space for more figures. The basic concept of g-distribution is discussed in detail with respect to ALOMAR data in section 3, 3.1.1, 3.1.2, and 3.2 (a total of approximately 6 pages including 3 figures). Also Appendix A summarizes some general properties of the g-function. Please keep in mind that the Thomas-Monograph-Paper is cited to reference correctly the first publication about*

*exponentially distributed (g-function) PMC data. During the last two decades many people use this g-function approach in analyses of PMC data.*

2. p. 3, line 8: This choice obscures the inter-annual variability in PMC behavior that has been demonstrated in many previous studies (e.g. Rong et al. [2014], DeLand and Thomas [2015], Fiedler et al. [2017]), which can represent a factor of two variation for the slope of the g-distribution at the latitude of ALOMAR. How does this averaging affect the applicability of the results derived later to any specific PMC season?
*Comment:*
*We had written at the end of section 2 (ALOMAR data description):*
*"In this paper we will analyze the climatology of all ice seasons from 2002 until 2016 merging all 15 seasons to one data record. Within this combined data set we then get a total number N of 8,597 observations which is sufficiently numerous in order to avoid too large statistical irregularities in a frequency histogram of the data."*

*Hence we treat the entire period 2002-2016 to get a larger number of data which improves the statistics of frequency rates. If one subdivides the total period into single seasons, we have to carefully test and analyze the single season numbers. Such a work is in preparation, and we think that these results will be presented in a subsequent publication.*

*Concerning your second comment that g-slopes sometimes differ from season to season which can represent a factor of two for the slope. Exactly, that point makes us suspicious. How do you know that in all these seasonal data analyses a g-approach is justified? We think this is not true, and therefore the slope-results should be considered with caution. Again we mention that we will address this task in a subsequent publication in the near future.*

3. p. 3, line 10: Is the three-color mode of operation used less often? Other papers discussing ALOMAR lidar measurements talk about 15-minute binning (e.g. Fiedler et al. [2017]), so I would actually expect many more individual profiles to be available from 15 years of data.
*Yes, this is true. One color measurements in the green laser line have been performed since 1997 and are much more frequent compared to the three-color measurements. Both techniques use integration times of 15 minutes. BUT for our DERIVATION of a new Z-pdf presented in this paper we wanted to use only three-color measurements because only those deliver SIMULTANEOUS measurements of max. backscatter, radius, number density, and ice water density at the height of beta-max. These simultaneous measurements are necessary to calculate correlation and regression plots that help to understand the physical meaning (power law dependence between parameters) of the new Z-pdf. Nevertheless, the Z-pdf can be constructed by a single data sample itself, see examples in section 5.3.*

4. p. 3, line 31: I'm not sure I understand how the standard deviation can be equal to the mean with a threshold of zero. Statistically, one sigma should not encompass all data smaller than the mean, but with the definition given in line 29, you are not allowing for negative values of x. So if (mean – st.dev.) = 0, where does (mean – 2*sigma) fall?
*Yes, this is true. Mean, variance, and ... can be always calculated for given distribution (Keyword: Moments of a pdf). See the mathematical derivation in Appendix A. Please keep in mind that all of our 4 ice parameters beta_max, r, n, and IMD are physical parameters with values larger than zero. Hence a beta-max value of zero does not exist, same argument holds for a case of negative beta-max. They do not exist!. The same is also true for n, IMD, and radius.*

*Please note, that this general property of a g-function, the mean is equal to the standard deviation, is used for our g-function test, see Eq 4.*

5. p. 5, line 6: I'm not sure that the term "obviously" is appropriate. The fit line in Figure 2a does go through the data, but the fluctuations between y = 40-90 look comparable in magnitude to those between x = 20-40 in Figure 1a. The roll off at y < 40 in Figure 2a is more significant to me, and it suggests that using a higher threshold (e.g. 40) would yield a satisfactory fit.
*Done, we delete 'Obviously' and replace the sentence with: We show in the following that the data points have not a precise linear shape.*
*A comment to a larger threshold of y>40: Such a tendency of choosing a special interval where linearity is more or less true can be often seen in literature. This is what we call "at least piecewise exponential shape", see introduction. Unfortunately, such an 'arbitrary' selection procedure is inadmissible in the sense of honest statistics.*
*Remember that we have chosen a threshold for beta_max of three, and the value of three is even a conservative estimate. Have a look at Fig 3a, the regression plot between beta-max (x) and ice mass density (y). You will see that the mean regression line (solid line) has a value of y=20 for x=3. Hence the y threshold is 20. And even for this case, a lot of data points have been canceled by the condition x>3 and y>20.*

6. p. 5, lines 17-18: Regarding "larger discrepancies", see comment #5.
*Yes, for ice radius and number density, exponential fits get even worse, see Fig 2c-f.*

7. p. 7, lines 3-4: MBS is a first-level measured quantity, whereas IMD, R, and n are derived based on various assumptions. Does this "failure" say something about the functional forms used to create the latter group of products?
*We think that this has no influence on the group of products.*

8. p. 8, lines 2-3: This figure uses data well below the previously defined fit threshold for both MBS and IMD. Does the result change if MBS > 3, IMD > 20 are required as specified for Figures 1 and 2?. What about IMD > 40, as suggested in comment #5?
*This figure uses ALL data that has been detected. As already said, a threshold of 3 is a conservative estimate for lidar sensitivity. Since summer 2002 there has been further development of the lidar system at almost every year. So, sensitivity became better and better. Does the result change if MBS > 3, IMD > 20 is used or other thresholds? We performed several numerical tests which show the following: regression points, mean, and median will always change when introducing different thresholds. But (c,d)- values change only slightly within a few percent (MBS > 3, IMD > 20).*

9. p. 13, lines 15-16: The first two derived threshold values are close to those given in Section 3.1.2. Is the third value a maximum?
*The third value is n_th = 662 cm-3, see discussion of n_ice in this section where we had given a description:*
*...The sample of ice number density shows a completely different behavior with a slope parameter that is negative with b= - 0.819. The physical meaning is that the parameter ice number density is negatively correlated with all other ice parameters. For example, large ice numbers n correspond to small ice radii, IMD and MBS values. As a consequence this leads to a threshold of n in the reverse direction, that is from large values to small values defined by n < n_th =662cm −3 ...*

10. p. 13, line 33: This statement seems to connect back to lines 23-24 on this page.
Isn't it circular reasoning to say that they agree?
*At lines 23-24 we discuss max. backscatter. At line 33 we discuss ice mass density (IMD).
Here we note that also numerical values of mean, median, and standard deviation for IMD
(and r and n ) agree almost perfectly. We think that these hints are justified.*

11. p. 14, lines 5-6: This statement is physically plausible for radius. It seems reasonable
for IMD which is proportional to $r^3$. Not sure about MBS, because it seems like
large density could overcome the dependence on r (but is this true if MBS is proportional
to $r^6$?).
*We calculated all correlations, and in fact this is the result.
As you say it is plausible for radius and IMD. IMD is proportional to $n* (radius^3)$.
MBS is proportional to $n* (radius^{5.8})$. You see the similarities. For that reason both pairs
(n,MBS) and (n,IMD) are negatively correlated.*

12. p. 18, lines 17-19: I'm still not convinced that the parameters derived from a multiseason
collection of data are valid to use for this type of "synthetic" data calculation
with a smaller subset of original IWC data, based on the previous comments about
interannual variations.
*The larger the data sample the better the statistics. The multi-season data sample describes
the general properties of the general frequency distribution without season-to-season
changes. As we described before we plan to write a second paper in the near future that
investigates the step from a decadal period to a single season.*

13. p. 20, lines 10-13: Is this statement saying that the uncertainty in the retrieval
assumptions is large enough to justify the difference in b? What level of agreement
would be needed for confidence?
*The conclusion is that the application of simplified assumptions used in the analytical
example (for example Gaussian distributed ice particles at the height of maximum
backscatter, constant ice particle number or spherical shape of ice particles) do not
reproduce a power constant d that results from the lidar observations. This is a systematic
difference that can't be explained by statistical errors.
We modified these sentences with:*
**The** *power constant (d=1.46) derived from the shape parameters b x and b y of the Z
distribution analysis of real ALOMAR IBS and IMD data is* **significantly** *different from the
power estimate (d=1.93) belonging to the analytical example that necessitates various
assumptions, e.g Gaussian distributed ice particles at the height of maximum backscatter,
constant ice particle number or spherical shape of ice particles. Hence, we conclude that the
determination of shape parameters b from a Z-distribution analysis* **of observational data**
*therefore provides …*

14. p. 20, lines 15-17: This goal would require quantitative answers to comment
**13 in order to be able to identify such changes. It also goes back to comment #2**
regarding the question of how these fits behave with different individual years of data,
and discussing how much noise increases with the reduction in the number of samples.
*As we mention this might be a future goal. We will try to address this point in a future paper.*

TYPOGRAPHICAL ERRORS

p. 5, lines 14-15: "unequal" could be "not equal to".
*Done*

p. 5, line 15: "unequal" could be "not equal to".
*Done*

p. 8, line 26: "allows to" should be "allows us to".
*Done*

p. 9, line 9: "particulary" should be "particularly".
*Done*

p. 14, line 8: "tale" should be "tail".
*Done*

p. 15, line 7: "Have in mind" could be "Please keep in mind".
*Done*

p. 16, line 11: "outcome" should be "outcomes".
*Done*

p. 24, line 12: "stimulus" could be "stimulating".
*Done*

---

## Author Comment (AC2) · 8 Nov 2018

**Response to Anonymous Referee #2**

Journal: ACP
Title: A new Description of Probability Density Distributions of Polar Mesospheric Clouds (PMC)
Author(s): Uwe Berger et al.
MS No.: acp-2018-642
MS Type: Research article
Special Issue: Layered phenomena in the mesopause region (ACP/AMT inter-journal SI)

*Introductory remarks:*
*We greatly appreciate the comments from the reviewer. We have taken his/her suggestions for improvements into account when preparing the revised version of the manuscript. In the following we respond to the reviewer's comments point by point.*
*We have marked the changes in the tracked version of the manuscript. Author responses are in italics. Line numbers refer to the first paper version. In the new tracked version deleted sequences are marked* red. *New text is marked in* blue.
*We want to thank the reviewer for the detailed review with many useful ideas and suggestions which, we think, have significantly increased the quality of the manuscript.*

GENERAL COMMENTS: This is an interesting and generally well written article dealing with probability density functions of various noctilucent cloud (NLC) parameters such as particle radius, cloud backscatter, ice particle density and ice mass density. While the backscatter is found to follow an exponential distribution, this is not the case for the other parameters considered. The NLC parameter database employed is based on the well-known Alomar LIDAR dataset. I do not have major objections against the publication of this article but ask the authors to consider the comments listed below. In addition, I have the following general comment: The LIDAR backscatter measurements, like all other optical measurements, are quite insensitive to particles with radii below a certain threshold. This is related to the finding that radii below about 20 nm are very infrequent in the data set, despite the fact that there are typically many more small particles than large particles. I think this aspect should be discussed in the paper, because it also (qualitatively) explains some of the differences between the PDFs of the different parameters.

*The Referee #2 is correct that optical instruments can not observe the smallest particles in PMC. However the cutoff at r > 20 nm is caused by the threshold of max beta > 3 and the fact that the limit the analysisto the peak of the layer.*
*In this paper we discuss 3 maximum backscatter signals that relate to a certain height within the PMC column (the height where backscatter maximizes). These are our measurements. From these 3 measurements we derive some more ice parameters as mean ice radius, standard deviation and ice mass density. These values refer exclusively to this specific height where the PMC shines brightest. Assuming a threshold in max beta of 3 produce the measured histograms in ice radius, ice mass density, and number density as shown in the plots. So it is not the question that the lidars are insensitive to small ice particles, a case that happens at the nucleation zone near the mesopause (87-90 km) where ice formation starts and up to several thousand ice particles exist (r<10 nm). But our lidar analysis takes into account only measurements from faint to strong brightness levels near the bottom (at height of beta max, 83 km) of the vertical PMC ice column.*

*We have extended the description of the retrieval of particle sizes in section 2 'Discription of ALOMAR lidar':*

*...After separation of the ice particle and molecular backscatter signal, we extract three vertical profiles of so-called backscatter ratios which are a measure of height dependent brightness of the ice cloud. From each backscatter height profile we estimate three maximum backscatter (MBS) values. We assume that at the altitude of MBS, typically located near 83 km, the actual shape of the ice particle distribution can be described by a Normal-distribution. Then we derive from the three measured MBS values the characteristics of the Normal-distribution with mean ice radius, ice number density and variance. Finally, we also estimate from these ice parameters the actual ice mass density (IMD) at the MBS height...*

This paper is well-written. Some suggestions and comments related to specific items are provided below.

SPECIFIC COMMENTS
Page 1, line 1 and line 15: "of Polar Mesospheric Clouds (PMC) and noctilucent clouds (NLC)."
This sounds like the two are different clouds. I suggest changing this sentence.
*Done: The expression NLC was considered redundant and was removed everywhere. Now, only the expression PMC is used throughout this paper.*

Page 1, line 5: "previously statistical methods" -> "previous statistical methods" or "previously used statistical methods"
*Done*

Page 1, line 6: "probability statistic"
Does "statistic" exist?
*Done: replaced by 'distributions'*

Page 1, line 12: "that facilitate" -> "that facilitates", because "facilitates" refers to "assessment",
right?
*Done*

Page 2, line 3: "many .. analysis" -> "many .. analyses"
*Done*

Page 2, line 8: "analysis have used" -> "analyses have used"
*Done*

Page 2, line 17 and line 18: "statistic" ?
*Done: statistics*

Page 2, line 31: "From each backscatter height profile we estimate a maximum backscatter (MBS) signal which corresponds to mean height of maximum brightness"
I don't fully understand this sentence. It mixes "signal" and "height" in a way, which makes it difficult to understand. Can you clarify, please?
*Done: ... From each backscatter height profile we estimate three maximum backscatter (MBS) values. We assume that at the altitude of MBS, typically located near 83 km, the actual shape of the ice particle distribution can be described by a Normal-distribution. Then we*

*derive from the three measured MBS values the characteristics of the Normal-distribution with mean ice radius, ice number density and variance. Finally, we also estimate from these ice parameters the actual ice mass density (IMD) at the MBS height.*

Page 3, line 15: "exponential distributed" -> "exponentially distributed"
*Done*

Page 3, line 30: "mode" is not a really frequently used term and I suggest briefly explaining it. It is explained on the next page and I suggest moving the explanation here.
*Done*

Page 4, line 19: "in a semi-logarithm scale". I suggest replacing this by "in a semilogarithmic diagram" (a scale can be linear or logarithmic, but not semi-logarithmic)
*Done*

Page 4, line 22: "Consequently, the relative error is rather small"
Please explain briefly how this relative error is determined.
*Done: The good quality of the fit is characterized by a small relative error of 6.5 % that is calculated as a sum of 100%· SUM(j,M) |E j −X j | for x >3 with theoretical exponential frequencies E j and normalized frequencies X j of data x per class j with a total of M classes.*

Page 5, line 5: " . . . as expected"
It's not entirely clear, what you consider to be expected. Do you expect that these other parameters also follow an exponential distribution or do you not? Please clarify.
*Done: 'as expected' has been deleted.*

Page 5, line 6: "in a semi-logarithmic scale" -> "in a semi-logarithmic diagram"
*Done*

Page 5, line 9: "significant smaller" -> "significantly smaller"
*Done*

Page 6, Caption Fig. 2, line 2: "least square fit" -> "least squares fit"
*Done*

Page 7, Fig. 3: I suggestion mentioning in the Figure caption what the dashed lines are.
*Done: The solid line shows the regression defined by regression points and corresponding (c,d)-values. Dashed lines result from regression analysis of y(x) : x ⇒ y and x(y) : x ⇒ y.*

Page 7, line 7: "Linearity between maximum backscatter (MBS) and ice mass density (IMD), ice radius r and ice number density n data is a necessary and sufficient condition that also IMD, r and n data samples are exponentially distributed"
I'm not sure you would really expect that MBS scales linearly with, e.g. radius. The intensity of the backscattered radiation does certainly not scale linearly with particle radius, right? Why should the maximum backscatter depend linearly on radius? If there are other indications etc. for that, please discuss. Considering that you use a power law to describe the relationship between two parameters, you don't really assume linearity, right? I think the term "linearity" should be replaced and then all is fine.

*This section is titled 'Test on linearity between maximum backscatter and ice mass density, ice radius, ice number density data'. Here we show in a first step that there exist no linearity between (MBS, r), (MBS, n), (MBS,IMD), and also other pairs as e.g. (n,r).*
*This gives a first theoretical hint why IMD,n, and r do not follow an exponential distribution as has been tested empirically in the section before (section 3.1.2.).*
*Or vice versus: In section 3.1.2 we plot exponential fit functions to all data samples (IBS,IMD,r, and n) and see for IMD,n, and r large statistical uncertainties for exponential fits that indicate that there is 'something wrong'!. Section 3.2 investigates this hypothesis in a different second way with the method of correlation and regression pairs.*
*Now we find a second reason for missing exponential fits of IMD,n, and r, since also linear regression fails. These are two independent different ways, and both saying that a g-function can't be the universal pdf for all four ice parameters.*

*This is the motivation why we introduce in the following sections a new z-pdf which now fits equally to ALL four parameters MBS,IMD,n, and r.*

Page 8, line 1: ". . . also relate to the half width of the angle"
Can you mention how the regression points "relate to" the half width of this angle? To me this is not obvious, but perhaps I'm missing something.
*Let us assume two data samples x and y. First, we calculate the means (x_mean, y_mean) and standard deviaitons s_x and s_y of x and y, plus the Pearson correlation coefficient R between x and y. Then a linear regression from x to y means to construct a linear fit with:*
*y = y_mean + R\*(s_y/s_x) \* (x - x_mean)     -> dashed line y(x)*

*A linear regression from y to x means to construct a linear fit with:*
*x = x_mean + R\*(s_x/s_y) \* (y - y_mean)*
*This equation can be transformed to y as y = y_mean + 1/R \* (s_y/s_x) \* (x - x_mean) and plotted as a second dashed line named x(y).*

*For R unequal to one the two regression lines are different. Then the best estimate is a 'mean' regression line as shown in Figure 3 (solid line) that cut the angle of the two classical regression lines (y(x),x(y))in half. The Formula is  y = y_mean + R\* s_y/s_x \*(x – x_mean) setting R=1.*

Page 8, line 5: "criteria" -> "criterion"
*Done*

Page 8, line 6: "which is far away from unity"
This is not surprising at all, because the backscatter does not scale linearly with radius.
But perhaps this is discussed below.
*At this point we simply show (prove) that, in fact, our lidar measurements of radius and max. backscatter do not scale linearly. Consequently, this non-linearity forces a modification of the exponential distribution (g-function) called Z-approach in our paper. The derivation and the characteristics of new Z-pdf and its connection to the former g-function are discussed in the following sections 4,5, and 6.*

Page 9, last line: Something is missing in this equation. "C" is not defined and it is neither required here. The integral can be explicitly evaluated, but I find that b>1 is a requirement for the integral being 1. Please check.

*Done: "C" was the constant of integration that has been deleted. Instead we use a vertical bar in order to show the integration boundaries of the antiderivative. This change has been also applied in the appendix.*

Page 10, line 10: Meaning of "Only, " at beginning of sentence not clear, at least to me. Without the comma it would make sense.
*Done*

Page 12, line 26: "should be here possible too" -> "should be possible here too"
*Done*

Page 13, line 15: Suggest replacing ", and resulting" by ", resulting in "
*Done*

Page 16, line 8: "in turns" -> "in turn"
*Done*

Page 16, line 12: "of an practical example" -> "of a practical example"
*Done*

Page 16, line 25: "Now the Z-distribution approach offers a more general possibility to derive artificial data samples without any knowledge of correlation and regression coefficients."
I'm not sure I fully agree with this statement, because information on the power law relationship between the two quantities is required, right?
*No: no information on the relationship between the two quantities is required. This is precisely the advantage of the new method. See text in section 6.1.*

The statement suggests (or may suggest) that no prior information on the relationship between the two quantities is needed, which is certainly not the case, because you assume that $a_r$ and $b_r$ are known.
*Sure this is true that $a_r$ and $b_r$ are assumed to be known. As we explain in detail in the text, the constants $a_r$ and $b_r$ are calculated from a Z-analysis of the sole radius data sample. No relationship between a pair of parameters is needed. We had already described these two assumptions one sentence later:*
*First, we assume that a data sample of x=MBS of number N exists and also its Z-distribution $Z(x, a_x, b_x)$ with $a_x = 0.140$ and $b_x = 0.931$ is well known, see Figure 5a. Secondly, we assume that we know a priori the form of the Z-distribution $Z(r, a_r, b_r)$ …*

Page 18, line 19: "of an satellite" -> "of a satellite"
*Done*

Page 19, line 7: I think "for $r_i > 37.5$ nm" should be "for $r_i < 37.5$ nm"
*Done, yes a typing error.*

Page 19, line 10: "signal result" -> "signals result"
*Done*

Page 19, line 11: "of an spherical ice particle" -> "of a spherical ice particle"
*Done*

Page 19, line 11/12: you assume a fixed relationship between LIDAR backscatter signal and particle radius. In reality, the power will decrease with increasing particle radius. For your estimation this certainly does not have to be considered, but it's perhaps worth mentioning.

*In reality, the power will sharply **increase** with increasing particle radius (~r\*\*5.8). We use this relationship explicitly in section 6.2 when calculating the integrals.*

Page 21, last line: "We present two numericalLY stable .."

*Done*

Page 21, line 6: "The Z-distribution approach offers a more general possibility to derive artificial data samples without any knowledge of correlation and regression coefficients" OK, but the approach requires a priori knowledge on the Z-distribution parameters, right?

*Comment: This sentence is related to a brief summary of section 6.1 (Construction of artificial data). In section 6.1 we state clearly for several times the assumption of a priori knowledge on the Z-distribution parameters.*